# CROSS-LAYER RETROSPECTIVE RETRIEVING VIA LAYER ATTENTION

**Yanwen Fang**[*1], **Yuxi Cai**[*1], **Jintai Chen**[2], **Jingyu Zhao**[1], **Guangjian Tian**[3], **Guodong Li**[†1]

[1]Department of Statistics & Actuarial Science, The University of Hong Kong
[2]College of Computer Science and Technology, Zhejiang University
[3]Huawei Noah's Ark Lab
{u3545683, caiyuxi, gladys17}@connect.hku.hk
jtigerchen@zju.edu.cn, Tian.Guangjian@huawei.com
gdli@hku.hk

## ABSTRACT

More and more evidence has shown that strengthening layer interactions can enhance the representation power of a deep neural network, while self-attention excels at learning interdependencies by retrieving query-activated information. Motivated by this, we devise a cross-layer attention mechanism, called multi-head recurrent layer attention (MRLA), that sends a query representation of the current layer to all previous layers to retrieve query-related information from different levels of receptive fields. A light-weighted version of MRLA is also proposed to reduce the quadratic computation cost. The proposed layer attention mechanism can enrich the representation power of many state-of-the-art vision networks, including CNNs and vision transformers. Its effectiveness has been extensively evaluated in image classification, object detection and instance segmentation tasks, where improvements can be consistently observed. For example, our MRLA can improve 1.6% Top-1 accuracy on ResNet-50, while only introducing 0.16M parameters and 0.07B FLOPs. Surprisingly, it can boost the performances by a large margin of 3-4% box AP and mask AP in dense prediction tasks. Our code is available at https://github.com/joyfang1106/MRLA.

## 1 INTRODUCTION

Growing evidence indicates that strengthening layer interactions can encourage the information flow of a deep neural network (He et al., 2016; Huang et al., 2017; Zhao et al., 2021). For example, in vision networks, the receptive fields are usually enlarged as layers are stacked. These hierarchical receptive fields play different roles in extracting features: local texture features are captured by small receptive fields, while global semantic features are captured by large receptive fields. Hence encouraging layer interactions can enhance the representation power of networks by combining different levels of features. Previous empirical studies also support the necessity of building interdependencies across layers. ResNet (He et al., 2016) proposed to add a skip connection between two consecutive layers. DenseNet (Huang et al., 2017) further reinforced layer interactions by making layers accessible to all subsequent layers within a stage. Recently, GLOM (Hinton, 2021) adopted an intensely interacted architecture that includes bottom-up, top-down, and same-level interactions, attempting to represent part-whole hierarchies in a neural network.

In the meantime, the attention mechanism has proven itself in learning interdependencies by retrieving query-activated information in deep neural networks. Current works about attention lay much emphasis on amplifying interactions within a layer (Hu et al., 2018; Woo et al., 2018; Dosovitskiy et al., 2021). They implement attention on channels, spatial locations, and patches; however, none of them consider attention on layers, which are actually the higher-level features of a network.

It is then natural to ask: "Can attention replicate its success in strengthening layer interactions?" This paper gives a positive answer. Specifically, starting from the vanilla attention, we first give a formal

---

* Authors contributed equally. † Correspondence to gdli@hku.hk.

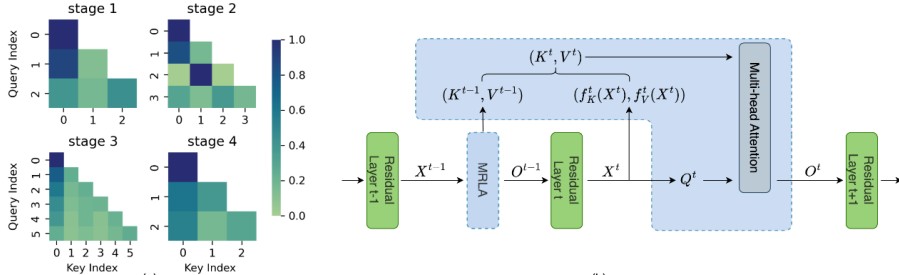

Figure 1: (a)Visualization of the layer attention scores from a randomly chosen head of MRLA in each stage of ResNet-50+MRLA model; (b) Schematic diagram of two consecutive layers with RLA.

definition of layer attention. Under this definition, a query representation of the current layer is sent to all previous layers to retrieve related information from hierarchical receptive fields. The resulting attention scores concretely depict the cross-layer dependencies, which also quantify the importance of hierarchical information to the query layer. Furthermore, utilizing the sequential structure of networks, we suggest a way to perform layer attention recurrently in Section 3.3 and call it recurrent layer attention (RLA). A multi-head design is naturally introduced to diversify representation subspaces, and hence comes multi-head RLA (MRLA). Figure 1(a) visualizes the layer attention scores yielded by MRLA at Eq. (6). Interestingly, most layers pay more attention to the first layer within the stage, verifying our motivation for retrospectively retrieving information.

Inheriting from the vanilla attention, MRLA has a quadratic complexity of $O(T^2)$, where $T$ is the depth of a network. When applied to very deep networks, this will incur a high computation cost and possibly the out-of-memory problem. To mitigate the issues, this paper makes an attempt to devise a light-weighted version of MRLA with linear complexity of $O(T)$. After imposing a linearized approximation, MRLA becomes more efficient and has a broader sphere of applications.

To our best knowledge, our work is the first attempt to systematically study cross-layer dependencies via attention. It is different from the information aggregation in DenseNet because the latter aggregates all previous layers' features in a channel-wise way regardless of which layer a feature comes from. OmniNet (Tay et al., 2021) and ACLA (Wang et al., 2022b) follow the same spirit as DensetNet. They allow each token from each layer to attend to all tokens from all previous layers. Essentially, both of them neglect the layer identity of each token. By contrast, we stand upon the layer attention to retrospectively retrieve query-related features from previous layers. Besides, to bypass the high computation cost, OmniNet divides the network into several partitions and inserts the omnidirectional attention block only after the last layer of each partition; and ACLA samples tokens with gates from each layer. Instead, our light-weighted version of MRLA can be easily applied to each layer.

The two versions of MRLA can improve many state-of-the-art (SOTA) vision networks, such as convolutional neural networks (CNNs) and vision transformers. We have conducted extensive experiments across various tasks, including image classification, object detection and instance segmentation. The experiment results show that our MRLA performs favorably against its counterparts. Especially in dense prediction tasks, it can outperform other SOTA networks by a large margin. The visualizations (see Appendix B.5) show that our MRLA can retrieve local texture features with positional information from previous layers, which may account for its remarkable success in dense prediction.

The main contributions of this paper are summarized below: (1) A novel layer attention, MRLA, is proposed to strengthen cross-layer interactions by retrieving query-related information from previous layers. (2) A light-weighted version of MRLA with linear complexity is further devised to make cross-layer attention feasible to more deep networks. (3) We show that MRLA is compatible with many networks, and validate its effectiveness across a broad range of tasks on benchmark datasets. (4) We investigate the important design elements of our MRLA block through an ablation study and provide guidelines for its applications on convolutional and transformer-based vision models.

## 2 RELATED WORK

**Layer Interaction**   Apart from the works mentioned above, other CNN-based and transformer-based models also put much effort into strengthening layer interactions. DIANet (Huang et al., 2020) utilized a parameter-sharing LSTM along the network depth to model the cross-channel relationships with the help of previous layers' information. CN-CNN (Guo et al., 2022) combined DIANet's LSTM and spatial and channel attention for feature fusion across layers. A similar RNN module was applied

along the network depth to recurrently aggregate layer-wise information in RLANet(Zhao et al., 2021). To distinguish it from our RLA, we rename the former as $\text{RLA}_g$ in the following. RealFormer (He et al., 2021) and EA-Transformer (Wang et al., 2021) both added attention scores in the previous layer to the current one, connecting the layers by residual attention. Bapna et al. (2018) modified the encoder-decoder layers by letting the decoders attend to all encoder layers. However, maintaining the features from all encoders suffers from a high memory cost, especially for high-dimensional features.

**Attention Mechanism in CNNs** CNNs have dominated vision tasks by serving as backbone networks in the past decades. Recently, attention mechanisms have been incorporated into CNNs with large receptive fields to capture long-range dependencies. SENet (Hu et al., 2018) and ECANet (Wang et al., 2020b) are two typical channel attention modules, which adaptively recalibrated channel-wise features by modelling the cross-channel dependencies. The Squeeze-and-Excitation (SE) block in SENet was later employed by the architectures of MobileNetV3 (Howard et al., 2019) and EfficientNet (Tan & Le, 2019). CBAM (Woo et al., 2018) first combined channel and spatial attention to emphasize meaningful features along the two principal dimensions. Pixel-level pairwise interactions across all spatial positions were captured by the non-local (NL) block in NLNet (Wang et al., 2018). GCNet (Cao et al., 2019) simplified the query-specific operation in the NL block to a query-independent one while maintaining the performances. CANet (Li et al., 2021) also extended the NL block for small object detection and integrated different layers' features by resizing and averaging. TDAM (Jaiswal et al., 2022) and BANet (Zhao et al., 2022) both perform joint attention on low- and high-level feature maps within a convolution block, which are considered as inner-layer attention in our paper.

**Transformer-based Vision Networks** Motivated by the success of Transformer (Vaswani et al., 2017) in Natural Language Processing (NLP), many researchers have applied transformer-based architectures to vision domains. The first is ViT (Dosovitskiy et al., 2021), which adapts a standard convolution-free Transformer to image classification by embedding an image into a sequence of patches. However, it relies on a large-scale pre-training to perform comparably with SOTA CNNs. This issue can be mitigated by introducing an inductive bias that the original Transformer lacks. DeiT (Touvron et al., 2021) adopted the knowledge distillation procedure to learn the inductive bias from a CNN teacher model. As a result, it only needs to be trained on a middle-size dataset, ImageNet-1K, and achieves competitive results as CNNs. CeiT (Yuan et al., 2021) introduced convolutions to the patch embedding process and the feed-forward network of ViT. Swin Transformer (Liu et al., 2021) employed a hierarchical design and a shifted-window strategy to imitate a CNN-based model.

## 3 LAYER ATTENTION AND RECURRENT LAYER ATTENTION

This section first recalls the mathematical formulation of self-attention. Then, it gives the definition of layer attention, and formulates the recurrent layer attention as well as its multi-head version.

### 3.1 REVISITING ATTENTION

Let $\boldsymbol{X} \in \mathbb{R}^{T \times D_{in}}$ be an input matrix consisting of $T$ tokens with $D_{in}$ dimensions each, and we consider a self-attention with an output matrix $\boldsymbol{O} \in \mathbb{R}^{T \times D_{out}}$. While in NLP each token corresponds to a word in a sentence, the same formalism can be applied to any sequence of $T$ discrete objects, e.g., pixels and feature maps.

The self-attention mechanism first derives the query, key and value matrices $\boldsymbol{Q}$, $\boldsymbol{K}$ and $\boldsymbol{V}$ by projecting $\boldsymbol{X}$ with linear transformations, i.e., $\boldsymbol{Q} = \boldsymbol{X}\boldsymbol{W}_Q$ with $\boldsymbol{W}_Q \in \mathbb{R}^{D_{in} \times D_k}$, $\boldsymbol{K} = \boldsymbol{X}\boldsymbol{W}_K$ with $\boldsymbol{W}_K \in \mathbb{R}^{D_{in} \times D_k}$, and $\boldsymbol{V} = \boldsymbol{X}\boldsymbol{W}_V$ with $\boldsymbol{W}_V \in \mathbb{R}^{D_{in} \times D_{out}}$. Then, the output is given by:

$$\boldsymbol{O} = \text{Self-Attention}(\boldsymbol{X}) := \text{softmax}(\frac{\boldsymbol{Q}\boldsymbol{K}^{\mathsf{T}}}{\sqrt{D_k}})\boldsymbol{V} = \boldsymbol{A}\boldsymbol{V},$$

where $\boldsymbol{A} = (a_{i,j})$ is a $T \times T$ matrix. Here we adopt NumPy-like notations: for a matrix $\boldsymbol{Y} \in \mathbb{R}^{I \times J}$, $\boldsymbol{Y}_{i,:}$, $\boldsymbol{Y}_{:,j}$, and $y_{i,j}$ are its $i$-th row, $j$-th column, and $(i,j)$-th element, respectively. Moreover, $[T]$ refers to the set of indices 1 to $T$. Then, a self-attention mechanism mapping any query token $t \in [T]$ from $D_{in}$ to $D_{out}$ dimensions can be formulated in an additive form:

$$\boldsymbol{O}_{t,:} = \boldsymbol{A}_{t,:}\boldsymbol{V} = \sum_{s=1}^{T} a_{t,s}\boldsymbol{V}_{s,:}. \tag{1}$$

## 3.2 LAYER ATTENTION

For a deep neural network, let $\boldsymbol{X}^t \in \mathbb{R}^{1 \times D}$ be the output feature of its $t$-th layer, where $t \in [T]$ and $T$ is the number of layers. We consider an attention mechanism with $\boldsymbol{X}^t$ attending to all previous layers and itself, i.e., the input matrix is $(\boldsymbol{X}^1, ..., \boldsymbol{X}^t) \in \mathbb{R}^{t \times D}$, and each $\boldsymbol{X}^s$ is treated as a token. Assuming $D_{in} = D_{out} = D$, we first derive the query, key and value for the $t$-th layer attention below,

$$\boldsymbol{Q}^t = f_Q^t(\boldsymbol{X}^t) \in \mathbb{R}^{1 \times D_k}, \tag{2}$$

$$\boldsymbol{K}^t = \text{Concat}[f_K^t(\boldsymbol{X}^1), ..., f_K^t(\boldsymbol{X}^t)] \quad \text{and} \quad \boldsymbol{V}^t = \text{Concat}[f_V^t(\boldsymbol{X}^1), ..., f_V^t(\boldsymbol{X}^t)], \tag{3}$$

where $\boldsymbol{K}^t \in \mathbb{R}^{t \times D_k}$ and $\boldsymbol{V}^t \in \mathbb{R}^{t \times D}$. Here, $f_Q^t$ denotes a function to extract current layer's information; $f_K^t$ and $f_V^t$ are functions that extract information from the 1st to $t$-th layers' features.

Denote the output of the $t$-th layer attention by $\boldsymbol{O}^t \in \mathbb{R}^{1 \times D}$, and then the layer attention with the $t$-th layer as the query is defined as follows:

$$\boldsymbol{O}^t = \boldsymbol{Q}^t(\boldsymbol{K}^t)^\mathsf{T}\boldsymbol{V}^t = \sum_{s=1}^{t} \boldsymbol{Q}^t(\boldsymbol{K}_{s,:}^t)^\mathsf{T}\boldsymbol{V}_{s,:}^t, \tag{4}$$

which has a similar additive form as in Eq. (1). The softmax and the scale factor $\sqrt{D_k}$ are omitted here for clarity since the normalization can be performed easily in practice. The proposed layer attention first depicts the dependencies between the $t$-th and $s$-th layers with the attention score $\boldsymbol{Q}^t(\boldsymbol{K}_{s,:}^t)^\mathsf{T}$, and then use it to reweight the transformed layer feature $\boldsymbol{V}_{s,:}^t$.

## 3.3 RECURRENT LAYER ATTENTION

In Eq. (3), the computation cost associated with $f_K^t$ and $f_V^t$ increases with $t$. Taking advantage of the sequential structure of layers, we make a natural simplification (see Appendix A.2 for details):

$$\boldsymbol{K}^t = \text{Concat}[f_K^1(\boldsymbol{X}^1), ..., f_K^t(\boldsymbol{X}^t)] \quad \text{and} \quad \boldsymbol{V}^t = \text{Concat}[f_V^1(\boldsymbol{X}^1), ..., f_V^t(\boldsymbol{X}^t)]. \tag{5}$$

The simplification allows the key and value matrices of the $t$-th layer to inherit from the preceding ones, i.e., $\boldsymbol{K}^t = \text{Concat}[\boldsymbol{K}^{t-1}, f_K^t(\boldsymbol{X}^t)]$ and $\boldsymbol{V}^t = \text{Concat}[\boldsymbol{V}^{t-1}, f_V^t(\boldsymbol{X}^t)]$, which avoids the redundancy induced by repeatedly deriving the keys and values for the same layer with different transformation functions. Based on this simplification, we can rewrite Eq. (4) into

$$\boldsymbol{O}^t = \sum_{s=1}^{t-1} \boldsymbol{Q}^t(\boldsymbol{K}_{s,:}^t)^\mathsf{T}\boldsymbol{V}_{s,:}^t + \boldsymbol{Q}^t(\boldsymbol{K}_{t,:}^t)^\mathsf{T}\boldsymbol{V}_{t,:}^t$$

$$= \sum_{s=1}^{t-1} \boldsymbol{Q}^t(\boldsymbol{K}_{s,:}^{t-1})^\mathsf{T}\boldsymbol{V}_{s,:}^{t-1} + \boldsymbol{Q}^t(\boldsymbol{K}_{t,:}^t)^\mathsf{T}\boldsymbol{V}_{t,:}^t = \boldsymbol{Q}^t(\boldsymbol{K}^{t-1})^\mathsf{T}\boldsymbol{V}^{t-1} + \boldsymbol{Q}^t(\boldsymbol{K}_{t,:}^t)^\mathsf{T}\boldsymbol{V}_{t,:}^t, \tag{6}$$

where the first term corresponds to the attention with the $t$-th layer sending the query to all previous layers. Compared to Eq. (4), $\boldsymbol{K}^t$ and $\boldsymbol{V}^t$ are now constructed in a recurrent way, i.e., $\boldsymbol{K}^{t-1}$ and $\boldsymbol{V}^{t-1}$ are reused. We therefore call Eq. (6) the recurrent layer attention (RLA), which significantly saves the computation cost. A schematic diagram is provided in Figure 1(b).

Nonetheless, RLA suffers from a quadratic complexity of the network depth since the first term in Eq. (6) still needs to be recomputed for each layer. Besides, for an image, the feature dimension $D$ of $\boldsymbol{V}^t$ is equal to $H \times W \times C$. For a very deep network with large $t$, the expanding size of $\boldsymbol{V}^t$ will result in the out-of-memory problem. Both of them limit the layer attention to a shallow or narrow network. Luckily, several techniques are available for linearizing the complexity of self-attention for the tokens within a layer. If workable at the layer level, they can be the tool that facilitates the layer attention to adapt for very deep networks under training time and memory constraints. We provide an example of linear RLA with the method proposed by Katharopoulos et al. (2020) in Sec.A.3 of Appendix.

Here we suggest another tailor-made method that utilizes an approximation of $\boldsymbol{Q}^t(\boldsymbol{K}^{t-1})^\mathsf{T}\boldsymbol{V}^{t-1}$ to linearize RLA. Denote the element-wise product by $\odot$, and then there exists a $\boldsymbol{\lambda}_q^t \in \mathbb{R}^{1 \times D_k}$ such that $\boldsymbol{Q}^t = \boldsymbol{\lambda}_q^t \odot \boldsymbol{Q}^{t-1}$. We speculate that query vectors at two consecutive layers have a similar pattern, i.e., $\boldsymbol{Q}^t$ is roughly proportional to $\boldsymbol{Q}^{t-1}$ or the elements of $\boldsymbol{\lambda}_q^t$ have similar values (see Figure 2(b) for empirical support). Consequently,

$$\boldsymbol{Q}^t(\boldsymbol{K}^{t-1})^\mathsf{T}\boldsymbol{V}^{t-1} = (\boldsymbol{\lambda}_q^t \odot \boldsymbol{Q}^{t-1})(\boldsymbol{K}^{t-1})^\mathsf{T}\boldsymbol{V}^{t-1}$$

$$\approx \boldsymbol{\lambda}_o^t \odot [\boldsymbol{Q}^{t-1}(\boldsymbol{K}^{t-1})^\mathsf{T}\boldsymbol{V}^{t-1}] = \boldsymbol{\lambda}_o^t \odot \boldsymbol{O}^{t-1}, \tag{7}$$

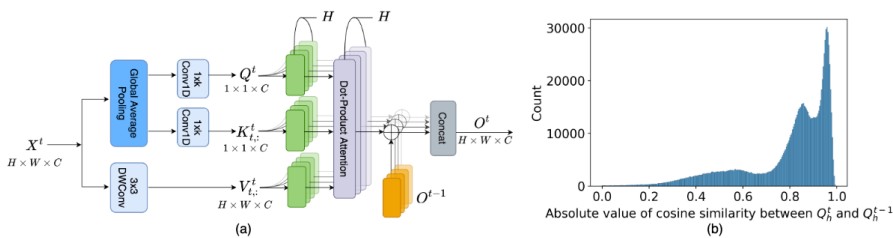

Figure 2: (a) Detailed operations in MRLA-light block with feature dimensions; (b) Absolute cosine similarity between queries from MRLA-base blocks of two consecutive layers.

where $\boldsymbol{\lambda}_o^t \in \mathbb{R}^{1 \times D}$ intrinsically depends on $\boldsymbol{Q}^t$, $\boldsymbol{Q}^{t-1}$ and $(\boldsymbol{K}^{t-1})^\mathsf{T} \boldsymbol{V}^{t-1}$, and we set $\boldsymbol{\lambda}_o^t$ as a learnable vector since its computation is complicated (see Appendix A.2 for the detailed explanation on the approximation above). The learnable vector can adaptively bridge the gap between $\boldsymbol{Q}^t$ and $\boldsymbol{Q}^{t-1}$. Note that the approximation in Eq. (7) becomes equivalency when the elements of $\boldsymbol{\lambda}_q^t$ are the same, i.e., $\boldsymbol{\lambda}_q^t = c\mathbf{1} \in \mathbb{R}^{1 \times D_k}$, and then $\boldsymbol{\lambda}_o^t = c\mathbf{1} \in \mathbb{R}^{1 \times D}$. Moreover, this approximation can be alleviated by the multi-head design in the next subsection.

With Eq. (7), an efficient and light-weighted RLA version with complexity $O(T)$ is suggested below:

$$\boldsymbol{O}^t = \boldsymbol{\lambda}_o^t \odot \boldsymbol{O}^{t-1} + \boldsymbol{Q}^t (\boldsymbol{K}_{t,:}^t)^\mathsf{T} \boldsymbol{V}_{t,:}^t = \sum_{l=0}^{t-1} \boldsymbol{\beta}_l \odot \left[ \boldsymbol{Q}^{t-l} (\boldsymbol{K}_{t-l,:}^{t-l})^\mathsf{T} \boldsymbol{V}_{t-l,:}^{t-l} \right], \tag{8}$$

where $\boldsymbol{\beta}_0 = \mathbf{1}$, and $\boldsymbol{\beta}_l = \boldsymbol{\lambda}_o^t \odot ... \odot \boldsymbol{\lambda}_o^{t-l+1}$ for $l \geq 1$. From the second equality, the extended RLA version is indeed a weighted average of past layers' features ($\boldsymbol{V}_{t-l,:}^{t-l}$). This is consistent with the broad definition of attention that many other attention mechanisms in computer vision use (Hu et al., 2018; Woo et al., 2018; Wang et al., 2020b). It is also interesting to observe that the first equality admits the form of a generalized residual connection, leading to an easier implementation in practice.

### 3.4 MULTI-HEAD RECURRENT LAYER ATTENTION

Motivated by the multi-head self-attention (MHSA) in Transformer, this section comes up with a multi-head recurrent layer attention (MRLA) to allow information from diverse representation subspaces. We split the terms in Eq. (6) and Eq. (8) into $H$ heads. Then, for head $h \in [H]$, RLA and its light-weighted version can be reformulated as

$$\boldsymbol{O}_h^t = \boldsymbol{Q}_h^t (\boldsymbol{K}_h^t)^\mathsf{T} \boldsymbol{V}_h^t \quad \text{and} \quad \boldsymbol{O}_h^t \approx \boldsymbol{\lambda}_{o,h}^t \odot \boldsymbol{O}_h^{t-1} + \boldsymbol{Q}_h^t (\boldsymbol{K}_{h[t,:]}^t)^\mathsf{T} \boldsymbol{V}_{h[t,:]}^t, \tag{9}$$

respectively, where $\boldsymbol{O}_h^t \in \mathbb{R}^{1 \times \frac{D}{H}}$ is the $h$-th head's output of MRLA. The final outputs are obtained by concatenation, i.e. $\boldsymbol{O}^t = \text{Concat}[\boldsymbol{O}_1^t, ..., \boldsymbol{O}_H^t]$. For convenience, we dub the MRLA and its light-weighted version as MRLA-base and MRLA-light, which are collectively referred to as MRLA. In contrast to Eq. (6) and Eq. (8) where a 3D image feature $\boldsymbol{V}_{t,:}^t$ is weighted by a scalar $\boldsymbol{Q}^t (\boldsymbol{K}_{t,:}^t)^\mathsf{T}$, the multi-head versions allow each layer's features to be adjusted by an enriched vector of length $H$, strengthening the representation power of the output features. In addition, for MRLA-light, as now the approximation in Eq. (7) is conducted within each head, i.e., $\boldsymbol{Q}_h^t (\boldsymbol{K}_h^{t-1})^\mathsf{T} \boldsymbol{V}_h^{t-1} \approx \boldsymbol{\lambda}_{o,h}^t \odot [\boldsymbol{Q}_h^{t-1} (\boldsymbol{K}_h^{t-1})^\mathsf{T} \boldsymbol{V}_h^{t-1}]$, it will become an equality as long as $\boldsymbol{\lambda}_{q,h}^t = c_h \mathbf{1} \in \mathbb{R}^{1 \times \frac{D_k}{H}}$, which is a much more relaxed requirement. In particular, when there are $D_k$ heads, i.e., $H = D_k$, the approximation in Eq. (7) always holds.

If layer attention is applied to small networks or the minimum computation, time and memory cost are not chased, we recommend both MRLA-base and MRLA-light as they can equivalently retrieve useful information from previous layers. In case one has to consider the training time and memory footprints, MRLA-light is preferred as an effecient version that we adapt for deeper networks.

## 4 APPLICATIONS OF MRLA IN VISION NETWORKS

Recent networks are formed by deep stacks of similar blocks (layers), and therefore a MRLA-base/light block can be inserted right after a building layer in a network.

**CNNs** Figure 2(a) illustrates the detailed block design of MRLA-light in CNNs (see that of MRLA-base in Figure 4 of Appendix). Given the output of the $t$-th CNN block $\boldsymbol{X}^t \in \mathbb{R}^{1 \times D}$, where $D = H \times W \times C$, we perform a global average pooling (GAP) to summarize the $t$-th layer's

information. Then two $1D$ convolutions (Conv1D) are used to extract the query $\boldsymbol{Q}^t$ and the key $\boldsymbol{K}_{t,:}^t$, whose kernel sizes are adaptively determined by the strategy in Wang et al. (2020b). A 3x3 depth-wise convolution (DWConv) is applied to directly get $\boldsymbol{V}_{t,:}^t$. Here, the two Conv1D (together with GAP($\cdot$)) and DWConv correspond to the $f_Q^t$, $f_K^t$ and $f_V^t$, respectively. We then divide the query, key and value into $H$ heads along the channels. The output of previous MRLA block, $\boldsymbol{O}^{t-1}$, is partitioned similarly. We set $\boldsymbol{\lambda}_o^t$ of size $C$ instead of $H \times W \times C$ in order to introduce fewer parameters, and it performs an element-wise product with $\boldsymbol{O}^{t-1}$ after being expanded. A sigmoid activation is added on the projection of query and key to scale the attention weights into $[0, 1]$.

**Vision Transformers**   Most of the block design is the same as in Figure 2(a) except for the following changes. The output of the $t$-th block of a vision transformer is $\boldsymbol{X}^t \in \mathbb{R}^{1 \times D}$, where $D = (N+1) \times C$, $N$ is the number of patches and $C$ is the embedding dimension. We first split $\boldsymbol{X}^t$ into patch tokens $\boldsymbol{X}_p^t \in \mathbb{R}^{N \times C}$ and a class token $\boldsymbol{X}_c^t \in \mathbb{R}^{1 \times C}$. Then the patch tokens that preserve spatial information are reshaped into $\boldsymbol{X}_p^t \in \mathbb{R}^{\sqrt{N} \times \sqrt{N} \times C}$ as the input of our MRLA. Only previous patch tokens of the last MRLA output $\boldsymbol{O}_p^{t-1}$ are brought into the MRLA block. Lastly, the patch tokens are reshaped to the initial dimension and concatenated with the class token as the next layer's input.

## 5   EXPERIMENTS

This section first evaluates our MRLAs by conducting experiments in image classification, object detection and instance segmentation. Then MRLA-light block is taken as an example to ablate its important design elements. All models are implemented by PyTorch toolkit on 4 V100 GPUs. More implementation details, results, comparisons and visualizations are provided in Appendix B.

### 5.1   IMAGENET CLASSIFICATION

We use the middle-sized ImageNet-1K dataset (Deng et al., 2009) directly. Our MRLAs are applied to the widely used ResNet (He et al., 2016) and the current SOTA EfficientNet (Tan & Le, 2019), which are two general ConvNet families. For vision transformers, DeiT (Touvron et al., 2021), CeiT (Yuan et al., 2021) and PVTv2 (Wang et al., 2022a) are considered. We compare our MRLAs with baselines and several SOTA attention methods using ResNet as a baseline model.

**Settings**   A hyperparameter $d_k$ is introduced to control the number of channels per MRLA head. We set $d_k = 32$, $8$ and $16$ for ResNet, EfficientNet and vision transformers, respectively. To train these vision networks with our MRLAs, we follow the same data augmentation and training strategies as in their original papers (He et al., 2016; Tan & Le, 2019; Touvron et al., 2021; Wang et al., 2022a).

**Results**   The performances of the different-sized SOTA networks with our MRLAs are reported in Table 1. For a fair comparison, the results of ResNets from torchvision are replicated. We first observe that MRLA-base and MRLA-light have comparable performances when added to relatively small networks, verifying that the approximation in MRLA-light does not sacrifice too much accuracy. The out-of-memory problem occurs when MRLA-base is applied to ResNet-101 with the same batch size. Therefore, MRLA-light is recommended for deeper networks if the efficiency and computational resources are taken into account, and it can perform slightly better probably because of the additional flexible learning vector. We next compare our MRLAs with other attention methods using ResNets as baselines. Results show that our MRLAs are superior to SENet, CBAM, $A^2$-Net, one NL, and ECA-Net. Especially among layer-interaction-related networks, our MRLAs outperform the DIANet and RLA$_g$-Net, all of which beat the DenseNet of similar model size. MRLAs are also as competitive as AA-Net and GCNet with lower model complexity or fewer parameters. For EfficientNets, MRLAs introduce about 0.01M and 0.02M more parameters, leading to 1.3% and 1.1% increases in Top-1 accuracy for EfficientNet-B0/B1, respectively. It is worth noting that the architecture of EfficientNets is obtained via a thorough neural architecture search which is hard to be further improved. Consistent improvements are also observed in the transformer-based models. Specifically, our MRLAs can achieve 1.2% and 1.0% gains in terms of Top-1 accuracy on DeiT-T and CeiT-T, while both introduce +0.03M parameters, and MRLA-light only increases +0.04B FLOPs. With CeiT-T, we also validate that the FLOPs induced by our MRLA are nearly linear to the input resolution (See Appendix B.1.2). We additionally supplement a fair comparison with BANet (Zhao et al., 2022) in Appendix B.

Table 1: Comparisons of single-crop accuracy on the ImageNet-1K validation set. † means the results are from torchvision toolkit. The bold fonts denote the best performances.

| Model Type | Model | Params | FLOPs | Input | Top-1 | Top-5 |
|---|---|---|---|---|---|---|
| CNNs | ResNet-50[†] (He et al., 2016) | 25.6 M | 4.1 B | 224 | 76.1 | 92.9 |
| | + SE (Hu et al., 2018) | 28.1 M | 4.1 B | 224 | 76.7 | 93.4 |
| | + CBAM (Woo et al., 2018) | 28.1 M | 4.2 B | 224 | 77.3 | 93.7 |
| | + $A^2$ (Chen et al., 2018) | 34.6 M | 7.0 B | 224 | 77.0 | 93.5 |
| | + AA (Bello et al., 2019) | 27.1 M | 4.5 B | 224 | 77.7 | 93.8 |
| | + 1 NL (Wang et al., 2018) | 29.0 M | 4.4 B | 224 | 77.2 | 93.5 |
| | + 1 GC (Cao et al., 2019) | 26.9 M | 4.1 B | 224 | 77.3 | 93.5 |
| | + all GC (Cao et al., 2019) | 29.4 M | 4.2 B | 224 | 77.7 | 93.7 |
| | + ECA (Wang et al., 2020b) | 25.6 M | 4.1 B | 224 | 77.5 | 93.7 |
| | + DIA (Huang et al., 2020) | 28.4 M | - | 224 | 77.2 | - |
| | + RLA$_g$ (Zhao et al., 2021) | 25.9 M | 4.5 B | 224 | 77.2 | 93.4 |
| | + MRLA-base (Ours) | 25.7 M | 4.6 B | 224 | **77.7** | **93.9** |
| | + MRLA-light (Ours) | 25.7 M | 4.2 B | 224 | **77.7** | 93.8 |
| | ResNet-101[†] (He et al., 2016) | 44.5 M | 7.8 B | 224 | 77.4 | 93.5 |
| | + SE (Hu et al., 2018) | 49.3 M | 7.8 B | 224 | 77.6 | 93.9 |
| | + CBAM (Woo et al., 2018) | 49.3 M | 7.9 B | 224 | 78.5 | 94.3 |
| | + AA (Bello et al., 2019) | 47.6 M | 8.6 B | 224 | 78.7 | 94.4 |
| | + ECA (Wang et al., 2020b) | 44.5 M | 7.8 B | 224 | 78.7 | 94.3 |
| | + RLA$_g$ (Zhao et al., 2021) | 45.0 M | 8.4 B | 224 | 78.5 | 94.2 |
| | + MRLA-light (Ours) | 44.9 M | 7.9 B | 224 | **78.7** | **94.4** |
| | ResNet-152 [†] (He et al., 2016) | 60.2 M | 11.6 B | 224 | 78.3 | 94.0 |
| | + SE (Hu et al., 2018) | 66.8 M | 11.6 B | 224 | 78.4 | 94.3 |
| | + ECA (Wang et al., 2020b) | 60.2 M | 11.6 B | 224 | 78.9 | 94.6 |
| | + RLA$_g$ (Zhao et al., 2021) | 60.8 M | 12.3 B | 224 | 78.8 | 94.4 |
| | + MRLA-light (Ours) | 60.7 M | 11.7 B | 224 | **79.1** | **94.6** |
| | EfficientNet-B0 (Tan & Le, 2019) | 5.3 M | 0.4 B | 224 | 77.1 | 93.3 |
| | + MRLA-base (Ours) | 5.3 M | 0.6 B | 224 | 78.3 | **94.1** |
| | + MRLA-light (Ours) | 5.3 M | 0.5 B | 224 | **78.4** | **94.1** |
| | EfficientNet-B1 (Tan & Le, 2019) | 7.8 M | 0.7 B | 240 | 79.1 | 94.4 |
| | + MRLA-base (Ours) | 7.8 M | 0.9 B | 240 | **80.2** | **95.3** |
| | + MRLA-light (Ours) | 7.8 M | 0.8 B | 240 | **80.2** | 95.2 |
| Vision Transformers | DeiT-T (Touvron et al., 2021) | 5.7 M | 1.2 B | 224 | 72.2 | 91.1 |
| | + MRLA-base (Ours) | 5.7 M | 1.4 B | 224 | **73.5** | **92.0** |
| | + MRLA-light (Ours) | 5.7 M | 1.2 B | 224 | 73.4 | 91.9 |
| | DeiT-S (Touvron et al., 2021) | 22.1 M | 4.5 B | 224 | 79.9 | 95.0 |
| | + MRLA-light (Ours) | 22.1 M | 4.6 B | 224 | **81.3** | **95.9** |
| | DeiT-B (Touvron et al., 2021) | 86.4 M | 16.8 B | 224 | 81.8 | 95.6 |
| | + MRLA-light (Ours) | 86.5 M | 16.9 B | 224 | **82.9** | **96.3** |
| | CeiT-T (Yuan et al., 2021) | 6.4 M | 1.4 B | 224 | 76.4 | 93.4 |
| | + MRLA-light (Ours) | 6.4 M | 1.4 B | 224 | **77.4** | **94.1** |
| | CeiT-T (Yuan et al., 2021) | 6.4 M | 5.1 B | 384 | 78.8 | 94.7 |
| | + MRLA-light (Ours) | 6.4 M | 5.1 B | 384 | **79.6** | **95.1** |
| | CeiT-S (Yuan et al., 2021) | 24.2 M | 4.8 B | 224 | 82.0 | 95.9 |
| | + MRLA-light (Ours) | 24.3 M | 4.9 B | 224 | **83.2** | **96.6** |
| | PVTv2-B0 (Wang et al., 2022a) | 3.4 M | 0.6 B | 224 | 70.5 | - |
| | + MRLA-base (Ours) | 3.4 M | 0.9 B | 224 | 71.4 | **90.7** |
| | + MRLA-light (Ours) | 3.4 M | 0.7 B | 224 | **71.5** | **90.7** |
| | PVTv2-B1 (Wang et al., 2022a) | 13.1 M | 2.3 B | 224 | 78.7 | - |
| | + MRLA-light (Ours) | 13.2 M | 2.4 B | 224 | **79.4** | **94.9** |

**Assumption Validation**    To validate the assumption we make in Eq. (7) that $\boldsymbol{Q}_h^t$ is roughly proportional to $\boldsymbol{Q}_h^{t-1}$ within each head, we compute the absolute value of the cosine similarity between them and visualize the histogram in Figure 2(b). Note that if the absolute cosine similarity approaches 1, the desire that the elements of $\boldsymbol{\lambda}_{q,h}^t$ have similar values is reached. The validation is conducted by randomly sampling 5 images from each class of the ImageNet validation set and then classifying these images with the trained ResNet-50+MRLA-base model. The query vectors from each head of all MRLA-base blocks are extracted except for those belonging to the first layer within each stage, as the first layer only attends to itself.

Table 2: Object detection results of different methods on COCO val2017. FLOPs are calculated on $1280 \times 800$ input. The bold fonts denote the best performances.

| Methods | Detectors | Params | $AP^{bb}$ | $AP^{bb}_{50}$ | $AP^{bb}_{75}$ | $AP^{bb}_{S}$ | $AP^{bb}_{M}$ | $AP^{bb}_{L}$ |
|---|---|---|---|---|---|---|---|---|
| ResNet-50 | | 41.53 M | 36.4 | 58.2 | 39.2 | 21.8 | 40.0 | 46.2 |
| + SE (Hu et al., 2018) | | 44.02 M | 37.7 | 60.1 | 40.9 | 22.9 | 41.9 | 48.2 |
| + ECA (Wang et al., 2020b) | | 41.53 M | 38.0 | 60.6 | 40.9 | 23.4 | 42.1 | 48.0 |
| + RLA$_g$ (Zhao et al., 2021) | | 41.79 M | 38.8 | 59.6 | 42.0 | 22.5 | 42.9 | 49.5 |
| + BA (Zhao et al., 2022) | | 44.66 M | 39.5 | 61.3 | 43.0 | **24.5** | 43.2 | 50.6 |
| + MRLA-base (Ours) | | 41.70 M | 40.1 | 61.3 | 43.8 | 24.0 | 43.9 | 52.4 |
| + MRLA-light (Ours) | Faster | 41.70 M | **40.4** | **61.5** | **44.0** | 24.2 | **44.1** | **52.7** |
| ResNet-101 | R-CNN | 60.52 M | 38.7 | 60.6 | 41.9 | 22.7 | 43.2 | 50.4 |
| + SE (Hu et al., 2018) | | 65.24 M | 39.6 | 62.0 | 43.1 | 23.7 | 44.0 | 51.4 |
| + ECA (Wang et al., 2020b) | | 60.52 M | 40.3 | 62.9 | 44.0 | 24.5 | 44.7 | 51.3 |
| + RLA$_g$ (Zhao et al., 2021) | | 60.92 M | 41.2 | 61.8 | 44.9 | 23.7 | 45.7 | 53.8 |
| + BA (Zhao et al., 2022) | | 66.44 M | 41.7 | **63.4** | 45.1 | 24.9 | 45.8 | 54.0 |
| + MRLA-light (Ours) | | 60.90 M | **42.0** | 63.1 | **45.7** | **25.0** | 45.8 | **55.4** |
| ResNet-50 | | 37.74 M | 35.6 | 55.5 | 38.2 | 20.0 | 39.6 | 46.8 |
| + SE (Hu et al., 2018) | | 40.23 M | 37.1 | 57.2 | 39.9 | 21.2 | 40.7 | 49.3 |
| + ECA (Wang et al., 2020b) | | 37.74 M | 37.3 | 57.7 | 39.6 | 21.9 | 41.3 | 48.9 |
| + RLA$_g$ (Zhao et al., 2021) | | 38.00 M | 37.9 | 57.0 | 40.8 | 22.0 | 41.7 | 49.2 |
| + MRLA-base (Ours) | | 37.92 M | 39.3 | 59.3 | 42.1 | 24.0 | 43.3 | 50.8 |
| + MRLA-light (Ours) | RetinaNet | 37.92 M | **39.6** | **59.7** | **42.4** | **24.1** | **43.6** | **51.2** |
| ResNet-101 | | 56.74 M | 37.7 | 57.5 | 40.4 | 21.1 | 42.2 | 49.5 |
| + SE (Hu et al., 2018) | | 61.45 M | 38.7 | 59.1 | 41.6 | 22.1 | 43.1 | 50.9 |
| + ECA (Wang et al., 2020b) | | 56.74 M | 39.1 | 59.9 | 41.8 | 22.8 | 43.4 | 50.6 |
| + RLA$_g$ (Zhao et al., 2021) | | 57.13 M | 40.3 | 59.8 | 43.5 | 24.2 | 43.8 | 52.7 |
| + MRLA-light (Ours) | | 57.12 M | **41.3** | **61.4** | **44.2** | **24.8** | **45.6** | **53.8** |

Table 3: Object detection and instance segmentation results of different methods using Mask R-CNN as a framework on COCO val2017. $AP^{bb}$ and $AP^{m}$ denote AP of bounding box and mask.

| Methods | Params | GFLOPs | $AP^{bb}$ | $AP^{bb}_{50}$ | $AP^{bb}_{75}$ | $AP^{m}$ | $AP^{m}_{50}$ | $AP^{m}_{75}$ |
|---|---|---|---|---|---|---|---|---|
| ResNet-50 | 44.18 M | 275.58 | 37.2 | 58.9 | 40.3 | 34.1 | 55.5 | 36.2 |
| + SE (Hu et al., 2018) | 46.67 M | 275.69 | 38.7 | 60.9 | 42.1 | 35.4 | 57.4 | 37.8 |
| + ECA (Wang et al., 2020b) | 44.18 M | 275.69 | 39.0 | 61.3 | 42.1 | 35.6 | 58.1 | 37.7 |
| + 1 NL (Wang et al., 2018) | 46.50 M | 288.70 | 38.0 | 59.8 | 41.0 | 34.7 | 56.7 | 36.6 |
| + GC (r16) (Cao et al., 2019) | 46.90 M | 279.60 | 39.4 | 61.6 | 42.4 | 35.7 | 58.4 | 37.6 |
| + GC (r4) (Cao et al., 2019) | 54.40 M | 279.60 | 39.9 | 62.2 | 42.9 | 36.2 | 58.7 | 38.3 |
| + RLA$_g$ (Zhao et al., 2021) | 44.43 M | 283.06 | 39.5 | 60.1 | 43.4 | 35.6 | 56.9 | 38.0 |
| + BA (Zhao et al., 2022) | 47.30 M | 261.98 | 40.5 | 61.7 | 44.2 | 36.6 | 58.7 | 38.6 |
| + MRLA-base (Ours) | 44.34 M | 289.49 | 40.9 | 62.1 | 44.8 | 36.9 | 58.8 | 39.3 |
| + MRLA-light (Ours) | 44.34 M | 276.93 | **41.2** | **62.3** | **45.1** | **37.1** | **59.1** | **39.6** |
| ResNet-101 | 63.17 M | 351.65 | 39.4 | 60.9 | 43.3 | 35.9 | 57.7 | 38.4 |
| + SE (Hu et al., 2018) | 67.89 M | 351.84 | 40.7 | 62.5 | 44.3 | 36.8 | 59.3 | 39.2 |
| + ECA (Wang et al., 2020b) | 63.17 M | 351.83 | 41.3 | 63.1 | 44.8 | 37.4 | 59.9 | 39.8 |
| + 1 NL (Wang et al., 2018) | 65.49 M | 364.77 | 40.8 | 63.1 | 44.5 | 37.1 | 59.9 | 39.2 |
| + GC (r16) (Cao et al., 2019) | 68.10 M | 354.30 | 41.1 | 63.6 | 45.0 | 37.4 | 60.1 | 39.6 |
| + GC (r4) (Cao et al., 2019) | 82.20 M | 354.30 | 41.7 | 63.7 | 45.5 | 37.6 | 60.5 | 39.8 |
| + RLA$_g$ (Zhao et al., 2021) | 63.56 M | 362.55 | 41.8 | 62.3 | 46.2 | 37.3 | 59.2 | 40.1 |
| + MRLA-light (Ours) | 63.54 M | 353.84 | **42.8** | **63.6** | **46.5** | **38.4** | **60.6** | **41.0** |

## 5.2 OBJECT DETECTION AND INSTANCE SEGMENTATION

This subsection validates the transferability and the generalization ability of our model in object detection and instance segmentation tasks using three typical object detection frameworks: Faster R-CNN (Ren et al., 2015), RetinaNet (Lin et al., 2017) and Mask R-CNN (He et al., 2017).

**Settings** All experiments are conducted on MS COCO 2017 (Lin et al., 2014), which contains 118K training, 5K validation and 20K test-dev images. All detectors are implemented by the open-source MMDetection toolkit (Chen et al., 2019), where the commonly used settings and 1x training schedule (Hu et al., 2018; Wang et al., 2020b; Zhao et al., 2021) are adopted.

**Results on Object Detection** Table 2 reports the results on COCO val2017 set by standard COCO metrics of Average Precision (AP). Surprisingly, our MRLAs boost the AP of the ResNet-50 and

Table 4: Ablation study on different variants using ResNet-50 as the baseline model.

| Model | Params | FLOPs | Top-1 | Model | Params | FLOPs | Top-1 |
|---|---|---|---|---|---|---|---|
| (a) MLA (Eq.(4)) | 25.7 M | 4.8 B | 77.6 | MRLA-light | 25.7 M | 4.2 B | **77.7** |
| MRLA-base | 25.7 M | 4.6 B | **77.7** | (e) - $\boldsymbol{\lambda}_o^t$=**1** | 25.7 M | 4.2 B | 77.1 |
| MRLA-light | 25.7 M | 4.2 B | **77.7** | - $d_k$=16 | 25.7 M | 4.2 B | 77.5 |
| (b) w/o $\boldsymbol{\lambda}_o^t \boldsymbol{O}^{t-1}$ | 25.7 M | 4.2 B | 77.0 | (f) - $d_k$=64 | 25.7 M | 4.2 B | 77.3 |
| (c) w/o DWConv2d | 25.6 M | 4.1 B | 77.4 | - $d_k$=1 (CRLA) | 25.7 M | 4.2 B | 77.2 |
| (d) w/ FC | 28.2 M | 4.2 B | 77.5 | (g) DWConv2d | 25.7 M | 4.2 B | 76.6 |

ResNet-101 by 4.0% and 3.5%, respectively. The improvements on other metrics are also significant, e.g., 3-4% on $AP_{50}$ and 4-6% on $AP_L$. In particular, the stricter criterion $AP_{75}$ can be boosted by 4-5%, suggesting a stronger localization performance. More excitingly, ResNet-50 with our MRLAs outperforms ResNet-101 by 2% on these detectors. Even when employing stronger backbones and detectors, the gains of our MRLAs are still substantial, demonstrating that our layer-level context modeling are complementary to the capacity of current models. Remarkably, they surpass all other models with large margins. Though $RLA_g$ and our MRLAs both strengthen layer interactions and thus bring the most performance gains, our layer attention outperforms layer aggregation in $RLA_g$.

**Comparisons Using Mask R-CNN** Table 3 shows MRLAs stand out with remarkable improvements on all the metrics. Especially, MRLA-light strikes a good balance between computational cost and notable gains. For example, it is superior to $RLA_g$ module and GC block, while using much lower model complexity. Even though BA-Net ('+ BA') utilizes better pre-trained weights obtained from more advanced ImageNet training settings, our approach still outperforms it in these tasks.

In summary, Tables 2 and 3 demonstrate that MRLAs can be well generalized to various tasks, among which they bring extraordinary benefits to dense prediction tasks. We make a reasonable conjecture that low-level features with positional information from local receptive fields are better preserved through layer interactions, leading to these notable improvements (see Figures 6 and 7 in Appendix).

## 5.3 ABLATION STUDY

**Different Variants of MRLA** Due to the limited resources, we experiment with ResNet-50 model on ImageNet. We first compare the multi-head layer attention (MLA) in Eq. (4) and MRLA-base in (a). Then we ablate the main components of MRLA-light to further identify their effects: (b) entirely without previous information; (c) without 3x3 DWConv; (d) replacing the 1D convolutions with fully-connected (FC) layers; (e) with the identity connection, i.e., $\boldsymbol{\lambda}_o^t$=**1**; (g) adding a 3x3 DWConv to each layer of the baseline model. We also compare different hyperparameters: (f) different $d_k$, including the special case of channel-wise recurrent layer attention (CRLA), i.e., $d_k = 1$.

**Results from Table 4** Comparing (a) and MRLA-light validates our approximation in Eq. (7). Since the 3x3 DWConv mainly controls the additional computation cost, we compare with inserting a 3x3 DWConv layer after each layer of the original ResNet-50. Comparing (c) and (g) with ours shows the improvement in accuracy is not fully caused by increasing model complexity. Then the necessities of retrieving previous layers' information and introducing $\boldsymbol{\lambda}_o^t$ are investigated. (b) indicates strengthening layers' interdependencies can boost the performance notably; while (e) shows MRLA plays a much more influential role than a simple residual connection. Using the FC layer in (d) may be unnecessary because it is comparable to Conv1D but introduces more parameters. (f) shows that our MRLA with different hyperparameters is still superior to the original network.

## 6 CONCLUSION AND FUTURE WORK

This paper focuses on strengthening layer interactions by retrospectively retrieving information via attention in deep neural networks. To this end, we propose a multi-head recurrent layer attention mechanism and its light-weighted version. Its potential has been well demonstrated by the applications on mainstream CNN and transformer models with benchmark datasets. Remarkably, MRLA exhibits a good generalization ability on object detection and instance segmentation tasks. Our first future work is to consider a comprehensive hyperparameter tuning and more designs that are still worth trying on CNNs and vision transformers, such as attempting other settings for the number of layer attention heads of each stage and using other transformation functions. Moreover, in terms of MRLA design at a higher level, it is also an interesting direction to adapt some other linearized attention mechanisms to a stack of layers.

**Reproducibility Statement** To supplement more explanations and experiments and ensure the reproducibility, we include the schematic diagram of multi-head layer attention, detailed inner structure design and pseudo codes of MRLA-base and MRLA-light in CNNs and vision transformers in Appendix A. Besides, more implementation details about baseline models with our MRLAs are documented in Appendix B.

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

APPENDIX

This appendix contains supplementary explanations and experiments to support the proposed layer attention and multi-head recurrent layer attention (MRLA). Appendix A supplements the arguments made in Section 3, including clarification on layer attention in Eq. (4), recurrent layer attention in Eq. (6) and (8) and another linearization of recurrent layer attention with the design in Katharopoulos et al. (2020). The pseudo codes of the two modules are also attached. Appendix B provides more experiment details, results, explanations for ablation study and visualizations of MRLA-base and MRLA-light in CNNs and vision transformers. Besides, we also attempt to support our motivation through experiment results.

## A MULTI-HEAD RECURRENT LAYER ATTENTION

A.1 discusses the differences between layer attention and other layer interaction related work. A.2 elaborates more on the simplification and approximation made in MRLA-base and MRLA-light. Besides, our attempt to linearize RLA with the recurrent method proposed in (Katharopoulos et al., 2020) is included in A.3, which is briefly mentioned in Section 3. A.4 explains more about the broad definition of attention. A.5 illustrates the detailed block design of MRLA-base and provides the pseudo codes of the both modules.

### A.1 LAYER INTERACTION RELATED WORK

Apart from the OmniNet (Tay et al., 2021) that we elaborate in Section 1, we would like to further compare our layer attention with other layer interaction related work, including DenseNet (Huang et al., 2017), DIA-Net (Huang et al., 2020) and $RLA_g$Net (Zhao et al., 2021). The empirical studies comparing them are included in Appendix B.1.3.

**Layer Attention** Figure 3 illustrates the layer attention proposed in Section 3.2. Here a residual layer refers to a CNN block or a Transformer block (including the multi-head attention layers and the feed-forward layer). For each layer, the feature maps of all preceding layers are used as its layer attention's inputs; and its own feature maps are also reused in all subsequent layer attention, which intrinsically has quadratic complexity.

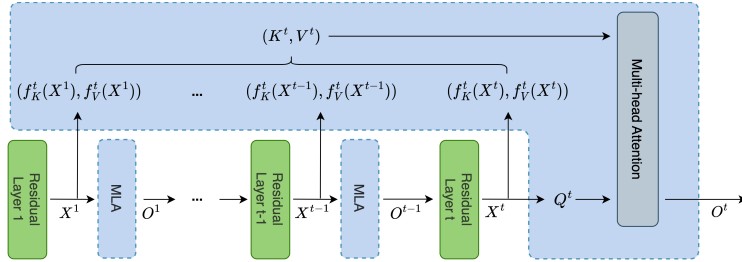

Figure 3: Schematic diagram of layer attention in Eq. (4).

**Comparison with DenseNet, DIANet, and $RLA_g$Net** Though DenseNet also extracts all previous layers' features and concatenates them before the Conv1x1, it is not an attention-based method. Moreover, it aggregates all previous layers' features in a channel-wise way regardless of which layer a feature comes from. It can be viewed as channel attention across layers. Instead, our layer attention models the features from the same layer in the same way via attention. $RLA_g$Net doesn't use attention, either. Finally, DIANet emphasizes the channel-wise recalibration of the current layer output, which is intrinsically channel attention instead of layer attention.

### A.2 DETAILS ABOUT SIMPLIFICATION IN EQ. (5) AND APPROXIMATION IN EQ. (8)

**Simplification in Eq. (5)** This simplification is weak and reasonable. In the vanilla self-attention, suppose we have $\boldsymbol{X} \in \mathbb{R}^{T \times D_{in}}$ where $T$ stands for the number of tokens and $D_{in}$ is the input

feature dimension. The vanilla self-attention derives the key and value matrix $\boldsymbol{K} \in \mathbb{R}^{T \times D_k}$ and $\boldsymbol{V} \in \mathbb{R}^{T \times D_{out}}$ by applying the transformation functions $F_K$ and $F_V$ respectively:

$$\boldsymbol{K} = F_K(\boldsymbol{X}) = F_K(\text{Concat}[\boldsymbol{X}_{1,:}, \boldsymbol{X}_{2,:}, ..., \boldsymbol{X}_{T,:}]),$$

and $\boldsymbol{V} = F_V(\boldsymbol{X}) = F_V(\text{Concat}[\boldsymbol{X}_{1,:}, \boldsymbol{X}_{2,:}, ..., \boldsymbol{X}_{T,:}])$. Importantly, both the transformation functions are linear projections, i.e., $F_K(\boldsymbol{X}) = XW_K$ and $F_V(\boldsymbol{X}) = XW_V$. They only operate on the feature dimension $D_{in}$ and thus

$$\boldsymbol{K} = \text{Concat}[F_K(\boldsymbol{X}_{1,:}), ..., F_K(\boldsymbol{X}_{T,:})] \quad \text{and} \quad \boldsymbol{V} = \text{Concat}[F_V(\boldsymbol{X}_{1,:}), ..., F_V(\boldsymbol{X}_{T,:})]. \tag{10}$$

Recall that in layer attention, we assume $D_{in} = D_{out} = D$ and denote the output feature of the $t$-th layer as $\boldsymbol{X}^t \in \mathbb{R}^{1 \times D}$. In the $t$-th layer attention, we derive the key and value $\boldsymbol{K}^t \in \mathbb{R}^{t \times D_k}$ and $\boldsymbol{V}^t \in \mathbb{R}^{t \times D}$ by applying the transformation functions $f_K^t$ and $f_V^t$ as in Eq.3:

$$\boldsymbol{K}^t = \text{Concat}[f_K^t(\boldsymbol{X}^1), ..., f_K^t(\boldsymbol{X}^t)] \quad \text{and} \quad \boldsymbol{V}^t = \text{Concat}[f_V^t(\boldsymbol{X}^1), ..., f_V^t(\boldsymbol{X}^t)].$$

In other words, we directly borrow the formulation in Eq.10 from the vanilla attention. Compared with the Eq.(5):

$$\boldsymbol{K}^t = \text{Concat}[f_K^1(\boldsymbol{X}^1), ..., f_K^t(\boldsymbol{X}^t)] \quad \text{and} \quad \boldsymbol{V}^t = \text{Concat}[f_V^1(\boldsymbol{X}^1), ..., f_V^t(\boldsymbol{X}^t)],$$

Eq. (3) has larger computation complexity. It is then natural to use the same transformation functions to avoid redundancy that is caused by repeatedly deriving the keys and values for the same layer with different transformation functions.

**Approximation in Eq. (8)** Denote the resulting matrix of $(\boldsymbol{K}^{t-1})^\mathsf{T}\boldsymbol{V}^{t-1}$ as $\boldsymbol{B}^{t-1} \in \mathbb{R}^{D_k \times D}$ with $\boldsymbol{b}_j^{t-1}$ being its $j$-th row vector. If the non-linear softmax function on the projection of query and key is omitted, then the first approximation in Eq. (7) is as follows:

$$\boldsymbol{Q}^t(\boldsymbol{K}^{t-1})^\mathsf{T}\boldsymbol{V}^{t-1} = (\boldsymbol{\lambda}_q^t \odot \boldsymbol{Q}^{t-1})[(\boldsymbol{K}^{t-1})^\mathsf{T}\boldsymbol{V}^{t-1}] \tag{11}$$

$$= (\boldsymbol{\lambda}_q^t \odot \boldsymbol{Q}^{t-1})\boldsymbol{B}^{t-1} \tag{12}$$

$$= \sum_{j=1}^{D_k} \lambda_{q,j}^t q_j^{t-1} \boldsymbol{b}_j^{t-1} \tag{13}$$

$$\approx c \sum_{j=1}^{D_k} q_j^{t-1} \boldsymbol{b}_j^{t-1} \tag{14}$$

$$= c[\boldsymbol{Q}^{t-1}(\boldsymbol{K}^{t-1})^\mathsf{T}\boldsymbol{V}^{t-1}], \tag{15}$$

where $\lambda_{q,j}^t$ and $q_j^{t-1}$ in (13) are the $j$-th element of $\boldsymbol{\lambda}_q^t \in \mathbb{R}^{1 \times D_k}$ and $\boldsymbol{Q}^{t-1} \in \mathbb{R}^{1 \times D_k}$, respectively. We approximate (13) with (14) by assuming all the elements of $\boldsymbol{\lambda}_q^t$ are the same. As mentioned in Section 4.1, the proposed multi-head design relaxes this condition by only requiring all the elements of $\boldsymbol{\lambda}_{q,h}^t \in \mathbb{R}^{1 \times \frac{D_k}{H}}$ are the same. We then generalize (15) to $\boldsymbol{\lambda}_o^t \odot [\boldsymbol{Q}^{t-1}(\boldsymbol{K}^{t-1})^\mathsf{T}\boldsymbol{V}^{t-1}]$ and set $\boldsymbol{\lambda}_o^t$ as learnable. This injects more flexibilities along the dimension $D$ and alleviates the effects of the simplification in (14).

## A.3    ANOTHER LINEARIZATION TECHNIQUE ON RLA

In addition to using a learnable parameter to bridge the previous layer attention output with the current one in Eq. (8), we have tried another technique to linearize the computation of layer attention. We can first rewrite the Eq. (6) with the softmax and the scale factor $\sqrt{D_k}$ as follows:

$$\boldsymbol{O}^t = \frac{\sum_{s=1}^t k(\boldsymbol{Q}^t, \boldsymbol{K}_{s,:}^t)\boldsymbol{V}_{s,:}^t}{\sum_{s=1}^t k(\boldsymbol{Q}^t, \boldsymbol{K}_{s,:}^t)},$$

where the kernel function $k(x, y) = \exp(\frac{x^\mathsf{T} y}{\sqrt{D_k}})$. Assuming that $k(x, y)$ can be approximated by another kernel with feature representation $\phi(\cdot)$, that is, $k(x, y) = \mathbb{E}[\phi(x)^\mathsf{T}\phi(y)]$ (Choromanski et al.,

2021). Then, the $t$-th layer attention output can be represented as:

$$
\begin{aligned}
\boldsymbol{O}^t &= \frac{\sum_{s=1}^{t} \phi(\boldsymbol{Q}^t)\phi(\boldsymbol{K}_{s,:}^t)^\mathsf{T}\boldsymbol{V}_{s,:}^t}{\sum_{s=1}^{t} \phi(\boldsymbol{Q}^t)\phi(\boldsymbol{K}_{s,:}^t)^\mathsf{T}} \\
&= \frac{\phi(\boldsymbol{Q}^t)\sum_{s=1}^{t} \phi(\boldsymbol{K}_{s,:}^t)^\mathsf{T}\boldsymbol{V}_{s,:}^t}{\phi(\boldsymbol{Q}^t)\sum_{s=1}^{t} \phi(\boldsymbol{K}_{s,:}^t)^\mathsf{T}}.
\end{aligned}
$$

The last equality holds because of the associative property of matrix multiplication. Adopting the linearization technique proposed by Katharopoulos et al. (2020), we introduce two variables:

$$
\boldsymbol{U}^t = \sum_{s=1}^{t} \phi(\boldsymbol{K}_{s,:}^t)^\mathsf{T}\boldsymbol{V}_{s,:}^t \quad \text{and} \quad \boldsymbol{Z}^t = \sum_{s=1}^{t} \phi(\boldsymbol{K}_{s,:}^t)^\mathsf{T},
$$

and simplify the computation as $\boldsymbol{O}^t = \frac{\phi(\boldsymbol{Q}^t)\boldsymbol{U}^t}{\phi(\boldsymbol{Q}^t)\boldsymbol{Z}^t}$. It is worth noting that $\boldsymbol{U}^t$ and $\boldsymbol{Z}^t$ can be computed from $\boldsymbol{U}^{t-1}$ and $\boldsymbol{Z}^{t-1}$ by

$$
\begin{aligned}
\boldsymbol{U}^t &= \boldsymbol{U}^{t-1} + \phi(\boldsymbol{K}_{t,:}^t)^\mathsf{T}\boldsymbol{V}_{t,:}^t \\
\boldsymbol{Z}^t &= \boldsymbol{Z}^{t-1} + \phi(\boldsymbol{K}_{t,:}^t)^\mathsf{T}
\end{aligned}
$$

This version of recurrent layer attention also results in a linear computation complexity with respect to the network depth. However, compared with Eq. (8), this attempt suffers from higher memory costs and a lower inference speed. Its multi-head implementation also performs poorer than our proposed MRLA-base/light in experiments. Therefore, we prefer weighting the previous layer attention output with a learnable parameter, which is a more efficient way to strengthen layer interactions in practice.

### A.4 MRLA-LIGHT IS CONSISTENT WITH THE BROAD DEFINITION OF ATTENTION

In Eq. (8), we have proven that

$$
\boldsymbol{O}^t = \sum_{l=0}^{t-1} \boldsymbol{\beta}_l \odot \left[\boldsymbol{Q}^{t-l}(\boldsymbol{K}_{t-l,:}^{t-l})^\mathsf{T}\boldsymbol{V}_{t-l,:}^{t-l}\right],
$$

which is a weighted average of past layers' information $\boldsymbol{V}_{t-l,:}^{t-l}$. This is consistent with the broad definition of attention that many other attention mechanisms in computer vision use.

More concretely, the existing channel attention or spatial attention mechanisms (SE(Hu et al., 2018), CBAM(Woo et al., 2018) and ECA(Wang et al., 2020b)) obtain the weights (similarities) of channels or pixels via learnable parameters (SE and ECA use two FC layers and a 1D convolution, while CBAM adopts a 2D convolution), and then scale the channels or pixels by the weights. This broader attention definition relaxes the requirement on the weights, allowing them not to fully depend on $\boldsymbol{X}$ but to be freely learnable.

Motivated by the above, the extended version of layer attention (MRLA-light) also scales different layers and takes the weighted average of them. It's worth mentioning that MRLA-light is different from SE in terms of SE recalibrating channels within a layer while MRLA-light makes adjustments across layers.

## A.5    MRLA BLOCKS AND PSEUDO CODES

Figure 4 illustrates the detailed operations in MRLA-base block with feature dimensions. Most of the inner structure designs are similar to those of MRLA-light block, except for concatenating previous keys and values instead of adding the output of previous MRLA block.

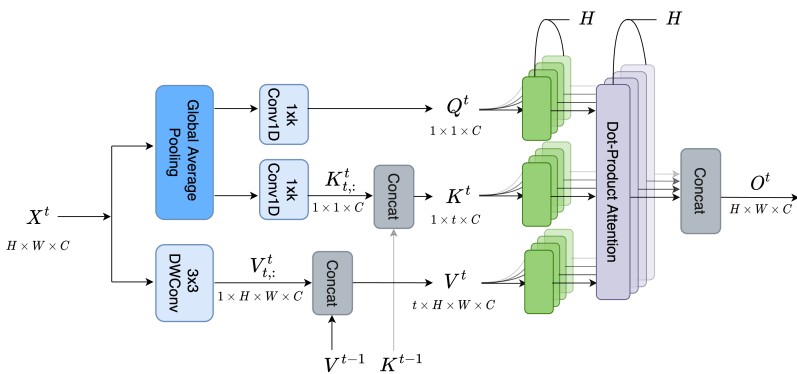

Figure 4: Detailed operations in MRLA-base block with feature dimensions.

**Pseudo Code**    Pseudo codes of MRLA-base's and MRLA-light's implementations in CNNs and vision transformers are given below.

---

**Algorithm 1** MRLA-base in CNNs

---

1: **Input:** Output of the $t$-th CNN block $\boldsymbol{X}^t \in \mathbb{R}^{1 \times H \times W \times C}$, $t-1$-th MRLA-base block's key and value $\boldsymbol{K}^{t-1} \in \mathbb{R}^{1 \times (t-1) \times C}$ and $\boldsymbol{V}^{t-1} \in \mathbb{R}^{1 \times (t-1) \times H \times W \times C}$, Number of heads $H$
2: **Output:** $\boldsymbol{O}^t \in \mathbb{R}^{1 \times H \times W \times C}$, $\boldsymbol{K}^t \in \mathbb{R}^{1 \times t \times C}$, $\boldsymbol{V}^t \in \mathbb{R}^{1 \times t \times H \times W \times C}$
3: // Summarize $\boldsymbol{X}^t$'s spatial information
4: $\boldsymbol{Y}^t \leftarrow \text{GAP}(\boldsymbol{X}^t) \in \mathbb{R}^{1 \times 1 \times C}$
5: // Derive the current layer's query, key and value via convolutions
6: $\boldsymbol{Q}^t \leftarrow \text{Conv1D}(\boldsymbol{Y}^t) \in \mathbb{R}^{1 \times 1 \times C}$
7: $\boldsymbol{K}_{t,:}^t \leftarrow \text{Conv1D}(\boldsymbol{Y}^t) \in \mathbb{R}^{1 \times 1 \times C}$
8: $\boldsymbol{V}_{t,:}^t \leftarrow \text{DWConv2D}(\boldsymbol{X}^t) \in \mathbb{R}^{1 \times 1 \times H \times W \times C}$
9: **If** $t = 1$ **then**
10: // First MRLA-base block
11:     $\boldsymbol{K}^t \leftarrow \boldsymbol{K}_{t,:}^t$
12:     $\boldsymbol{V}^t \leftarrow \boldsymbol{V}_{t,:}^t$
13: **else**
14: // Concatenate with the previous key and value
15:     $\boldsymbol{K}^t \leftarrow \text{Concat}[\boldsymbol{K}^{t-1}, \boldsymbol{K}_{t,:}^t] \in \mathbb{R}^{1 \times t \times C}$
16:     $\boldsymbol{V}^t \leftarrow \text{Concat}[\boldsymbol{V}^{t-1}, \boldsymbol{V}_{t,:}^t] \in \mathbb{R}^{1 \times t \times H \times W \times C}$
17: **End if**
18: $\boldsymbol{O}^t \leftarrow \text{Multi-head Attention}(\boldsymbol{Q}^t, \boldsymbol{K}^t, \boldsymbol{V}^t)$ where the number of heads is $H$

---

## B    EXPERIMENTS

Due to the limited space in the main paper, we provide more experimental settings and discussions in this section which are organized as follows. We first provide the implementation details of ImageNet classification and more comparisons with other SOTA networks in Appendix B.1. We also compare the model complexity and memory cost of MLA and MRLAs in Appendix B.2. Next, some implementation details and results of object detection and instance segmentation on COCO are included in Appendix B.3. Then the ablation study of a vision transformer (DeiT) is given in

---

**Algorithm 2** MRLA-base in Vision Transformers

---

1: **Input:** Output of the $t$-th transformer block $\boldsymbol{X}^t \in \mathbb{R}^{1 \times (N+1) \times C}$, $t-1$-th MRLA-base block's key and value $\boldsymbol{K}^{t-1} \in \mathbb{R}^{1 \times (t-1) \times C}$ and $\boldsymbol{V}^{t-1} \in \mathbb{R}^{1 \times (t-1) \times \sqrt{N} \times \sqrt{N} \times C}$, Number of heads $H$

2: **Output:** $\boldsymbol{O}^t \in \mathbb{R}^{1 \times (N+1) \times C}$, $\boldsymbol{K}^t \in \mathbb{R}^{1 \times t \times C}$, $\boldsymbol{V}^t \in \mathbb{R}^{1 \times t \times \sqrt{N} \times \sqrt{N} \times C}$

3: // Split into the class token and patch tokens

4: $\boldsymbol{X}_c^t \in \mathbb{R}^{1 \times 1 \times C}$, $\boldsymbol{X}_p^t \in \mathbb{R}^{1 \times N \times C} \leftarrow \text{Split}(\boldsymbol{X}^t)$

5: //Reshape

6: $\boldsymbol{X}_p^t \leftarrow \text{Reshape}(\boldsymbol{X}_p^t) \in \mathbb{R}^{1 \times \sqrt{N} \times \sqrt{N} \times C}$

7: // Summarize $\boldsymbol{X}_p^t$'s spatial information

8: $\boldsymbol{Y}_p^t \leftarrow \text{GAP}(\boldsymbol{X}_p^t) \in \mathbb{R}^{1 \times 1 \times C}$

9: // Derive the current layer's query, key and value via convolutions

10: $\boldsymbol{Q}^t \leftarrow \text{Conv1D}(\boldsymbol{Y}_p^t) \in \mathbb{R}^{1 \times 1 \times C}$

11: $\boldsymbol{K}_{t,:}^t \leftarrow \text{Conv1D}(\boldsymbol{Y}_p^t) \in \mathbb{R}^{1 \times 1 \times C}$

12: $\boldsymbol{V}_{t,:}^t \leftarrow \text{DWConv2D}(\boldsymbol{X}_p^t) \in \mathbb{R}^{1 \times 1 \times \sqrt{N} \times \sqrt{N} \times C}$

13: **If** $t = 1$ **then**

14: // First MRLA-base block

15:  $\boldsymbol{K}^t \leftarrow \boldsymbol{K}_{t,:}^t$

16:  $\boldsymbol{V}^t \leftarrow \boldsymbol{V}_{t,:}^t$

17: **else**

18: // Concatenate with the previous key and value

19:  $\boldsymbol{K}^t \leftarrow \text{Concat}[\boldsymbol{K}^{t-1}, \boldsymbol{K}_{t,:}^t] \in \mathbb{R}^{1 \times t \times C}$

20:  $\boldsymbol{V}^t \leftarrow \text{Concat}[\boldsymbol{V}^{t-1}, \boldsymbol{V}_{t,:}^t] \in \mathbb{R}^{1 \times t \times \sqrt{N} \times \sqrt{N} \times C}$

21: **End if**

22: $\boldsymbol{O}_p^t \leftarrow \text{Multi-head Attention}(\boldsymbol{Q}^t, \boldsymbol{K}^t, \boldsymbol{V}^t)$ where the number of heads is $H$

23: // Reshape to the original dimension

24: $\boldsymbol{O}_p^t \leftarrow \text{Reshape}(\boldsymbol{O}_p^t) \in \mathbb{R}^{1 \times N \times C}$

25: $\boldsymbol{O}^t \leftarrow \text{Concat}[\boldsymbol{X}_c^t, \boldsymbol{O}_p^t]$

---

**Algorithm 3** MRLA-light in CNNs

---

1: **Input:** Output of the $t$-th CNN block $\boldsymbol{X}^t \in \mathbb{R}^{1 \times H \times W \times C}$, $t-1$-th MRLA-light block's output $\boldsymbol{O}^{t-1} \in \mathbb{R}^{1 \times H \times W \times C}$, Number of heads $H$, Learnable parameter $\boldsymbol{\lambda}_o^t \in \mathbb{R}^{1 \times C}$

2: **Output:** $\boldsymbol{O}^t \in \mathbb{R}^{1 \times H \times W \times C}$

3: // Summarize $\boldsymbol{X}^t$'s spatial information

4: $\boldsymbol{Y}^t \leftarrow \text{GAP}(\boldsymbol{X}^t) \in \mathbb{R}^{1 \times 1 \times C}$

5: // Derive the current layer's query, key and value via convolutions

6: $\boldsymbol{Q}^t \leftarrow \text{Conv1D}(\boldsymbol{Y}^t) \in \mathbb{R}^{1 \times 1 \times C}$

7: $\boldsymbol{K}_{t,:}^t \leftarrow \text{Conv1D}(\boldsymbol{Y}^t) \in \mathbb{R}^{1 \times 1 \times C}$

8: $\boldsymbol{V}_{t,:}^t \leftarrow \text{DWConv2D}(\boldsymbol{X}^t) \in \mathbb{R}^{1 \times 1 \times H \times W \times C}$

9: $\tilde{\boldsymbol{O}}^t \leftarrow \text{Multi-head Attention}(\boldsymbol{Q}^t, \boldsymbol{K}_{t,:}^t, \boldsymbol{V}_{t,:}^t)$ where the number of heads is $H$

10: $\boldsymbol{O}^t \leftarrow \text{Expand}(\boldsymbol{\lambda}_o^t) \odot \boldsymbol{O}^{t-1} + \tilde{\boldsymbol{O}}^t$

---

---

**Algorithm 4** MRLA-light in Vision Transformers

---

1: **Input:** Output of the $t$-th transformer block $\boldsymbol{X}^t \in \mathbb{R}^{1 \times (N+1) \times C}$, $t-1$-th MRLA-light block's output $\boldsymbol{O}^{t-1} \in \mathbb{R}^{1 \times (N+1) \times C}$, Number of heads $H$, Learnable parameter $\boldsymbol{\lambda}_o^t \in \mathbb{R}^{1 \times C}$
2: **Output:** $\boldsymbol{O}^t \in \mathbb{R}^{1 \times (N+1) \times C}$
3: // Split into the class token and patch tokens
4: $\boldsymbol{X}_c^t \in \mathbb{R}^{1 \times 1 \times C}, \boldsymbol{X}_p^t \in \mathbb{R}^{1 \times N \times C} \leftarrow \text{Split}(\boldsymbol{X}^t)$
5: $\boldsymbol{O}_c^{t-1} \in \mathbb{R}^{1 \times 1 \times C}, \boldsymbol{O}_p^{t-1} \in \mathbb{R}^{1 \times N \times C} \leftarrow \text{Split}(\boldsymbol{O}^{t-1})$
6: //Reshape
7: $\boldsymbol{X}_p^t \leftarrow \text{Reshape}(\boldsymbol{X}_p^t) \in \mathbb{R}^{1 \times \sqrt{N} \times \sqrt{N} \times C}$
8: // Summarize $\boldsymbol{X}_p^t$'s spatial information
9: $\boldsymbol{Y}_p^t \leftarrow \text{GAP}(\boldsymbol{X}_p^t) \in \mathbb{R}^{1 \times 1 \times C}$
10: // Derive the current layer's query, key and value via convolutions
11: $\boldsymbol{Q}^t \leftarrow \text{Conv1D}(\boldsymbol{Y}_p^t) \in \mathbb{R}^{1 \times 1 \times C}$
12: $\boldsymbol{K}_{t,:}^t \leftarrow \text{Conv1D}(\boldsymbol{Y}_p^t) \in \mathbb{R}^{1 \times 1 \times C}$
13: $\boldsymbol{V}_{t,:}^t \leftarrow \text{GELU}(\text{DWConv2D}(\boldsymbol{X}_p^t)) \in \mathbb{R}^{1 \times 1 \times \sqrt{N} \times \sqrt{N} \times C}$
14: $\tilde{\boldsymbol{O}}_p^t \leftarrow \text{Multi-head Attention}(\boldsymbol{Q}^t, \boldsymbol{K}_{t,:}^t, \boldsymbol{V}_{t,:}^t)$ where the number of heads is $H$
15: // Reshape to the original dimension
16: $\tilde{\boldsymbol{O}}_p^t \leftarrow \text{Reshape}(\tilde{\boldsymbol{O}}_p^t) \in \mathbb{R}^{1 \times N \times C}$
17: $\boldsymbol{O}_p^t \leftarrow \text{Expand}(\boldsymbol{\lambda}_o^t) \odot \boldsymbol{O}_p^{t-1} + \tilde{\boldsymbol{O}}_p^t$
18: $\boldsymbol{O}^t \leftarrow \text{Concat}[\boldsymbol{X}_c^t, \boldsymbol{O}_p^t]$

---

Appendix B.4. Finally, visualizations of the feature maps/attention maps in ResNet50/DeiT and our MRLA counterparts are shown in Appendix B.5. All experiments are implemented on four Tesla V100 GPUs (32GB).

## B.1 IMAGENET CLASSIFICATION

### B.1.1 IMPLEMENTATION DETAILS

**ResNet** For training ResNets with our MRLA, we follow exactly the same data augmentation and hyper-parameter settings in original ResNet. Specifically, the input images are randomly cropped to $224 \times 224$ with random horizontal flipping. The networks are trained from scratch using SGD with momentum of 0.9, weight decay of 1e-4, and a mini-batch size of 256. The models are trained within 100 epochs by setting the initial learning rate to 0.1, which is decreased by a factor of 10 per 30 epochs. Since the data augmentation and training settings used in ResNet are outdated, which are not as powerful as those used by other networks, strengthening layer interactions leads to overfitting on ResNet. Pretraining on a larger dataset and using extra training settings can be an option; however, as most of our baseline models and the above attention models are not pretrained on larger datasets, these measures will result in an unfair comparison. Hence we use a more efficient strategy: applying stochastic depth (Huang et al., 2016) with survival probability of 0.8 only on our MRLA, the effects of which will be discussed in the ablation study later.

**EfficientNet** For training EfficientNet with our MRLA, we follow the settings in EfficientNet. Specifically, networks are trained within 350 epochs using RMSProp optimizer with momentum of 0.9, decay of 0.9, batch norm momentum of 0.99, weight decay of 4e-5 and mini-batch size of 4096. The initial learning rate is set to 0.256, and is decayed by 0.97 every 2.4 epochs. Since our computational resources hardly support the original batch size, we linearly scale the initial learning rate and the batch size to 0.048 and 768, respectively. We also use AutoAugment (Cubuk et al., 2019), stochastic depth (Huang et al., 2016) with survival probability 0.8 and dropout (Srivastava et al., 2014) ratio 0.2 for EfficientNet-B0 and EfficientNet-B1. Our MRLA shares the same stochastic depth with the building layer of EfficientNet since it is natural to drop the MRLA block if the corresponding layer is dropped. We implement EfficientNets with these training tricks by pytorch-image-models (timm) toolkit (Wightman, 2019) on 2x V100 GPUs. [1]

---

[1]License: Apache License 2.0

**DeiT, CeiT and PVTv2** We adopt the same training and augmentation strategy as that in DeiT. All models are trained for 300 epochs using the AdamW optimizer with weight decay of 0.05. We use the cosine learning rate schedule and set the initial learning rate as 0.001 with batch size of 1024. Five epochs are used to gradually warm up the learning rate at the beginning of the training. We apply RandAugment (Cubuk et al., 2020), repeated augmentation (Hoffer et al., 2020), label smoothing (Szegedy et al., 2016) with $\epsilon = 0.1$, Mixup (Zhang et al., 2017) with 0.8 probability, Cutmix (Yun et al., 2019) with 1.0 probability and random erasing (Zhong et al., 2020) with 0.25 probability. Similarly, our MRLA shares the same probability of the stochastic depth with the MHSA and FFN layers of DeiT/CeiT/PVTv2. xNote that since PVTv2 is a multi-stage architecture and the size of feature maps differs across the stage, we perform layer attention within each stage. DeiT and CeiT have 12 layers in total and we partition them into three stages, each with four layers, and apply the MRLA within the stage.

### B.1.2 MODEL COMPLEXITY WITH RESPECT TO INPUT RESOLUTION

Figure 5 visualizes the FLOPs induced by MRLA-light with respect to the input resolution. We compute the FLOPs of the baseline CeiT-T and our MRLA-light counterpart and then derive their differences under various settings of input resolution. It can be observed that the complexity of MRLA-light is linear to the input resolution.

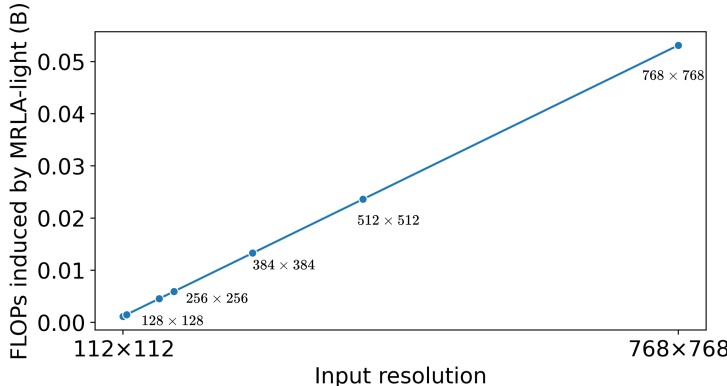

Figure 5: The FLOPs induced by MRLA-light with respect to input resolution.

### B.1.3 COMPARISONS WITH RELEVANT NETWORKS

**Layer-interation-related Networks** We first compare our MRLAs with DenseNet(Huang et al., 2017), DIANet (Huang et al., 2020) and RLA$_g$Net (Zhao et al., 2021) empirically. Their comparisons on the ImageNet-1K validation set are given in Table 5. Our MRLAs outperform the DIANet and RLA$_g$Net, all of which beat the DenseNet of similar model size.

**Other Relevant Networks** TDAM (Jaiswal et al., 2022) and BA-Net (Zhao et al., 2022) adopted different implemental settings from ours in the training of ImageNet. For the baseline model, we used the results from the torchvision toolkit while TDAM utilized those from the pytorch-image-models (timm). Note that the latter implementation includes advanced design settings (e.g., three 3x3 Convs instead of a 7x7 Conv) and training tricks (e.g., cosine learning schedule and label smoothing) to improve the performance of ResNets. And BA-Net applied cosine learning schedule and label smoothing in their training process. Therefore, it would be unfair to directly evaluate the performances between TDAM, BA-Net and MRLAs using the current results. We reproduced the performance of BA-Net with our training settings and train our model with the settings used in BA-Net. The results are given in Table 6, and we also compare the performances though we do not train our model by using the better pre-trained weights in object detection and instance segmentation tasks.

Table 5: Performances of layer-interation-related networks on the ImageNet-1K validation set.

| Model | Params | FLOPs | Top-1 | Top-5 |
|---|---|---|---|---|
| ResNet-50 | 25.6 M | 4.1 B | 76.1 | 92.9 |
| + DIA (Huang et al., 2020) | 28.4 M | - | 77.2 | - |
| + RLA$_g$ (Zhao et al., 2021) | 25.9 M | 4.5 B | 77.2 | 93.4 |
| + MRLA-base (Ours) | 25.7 M | 4.6 B | **77.7** | **93.9** |
| + MRLA-light (Ours) | 25.7 M | 4.2 B | **77.7** | 93.8 |
| ResNet-101 | 44.5 M | 7.8 B | 77.4 | 93.5 |
| + RLA$_g$ (Zhao et al., 2021) | 45.0 M | 8.4 B | 78.5 | 94.2 |
| + MRLA-light (Ours) | 44.9 M | 7.9 B | **78.7** | **94.4** |
| DenseNet-161 (k=48) (Huang et al., 2017) | 27.4 M | 7.9 B | 77.7 | 93.8 |
| DenseNet-264 (k=32) (Huang et al., 2017) | 31.8 M | 5.9 B | 77.9 | 93.8 |

Table 6: Comparisons of parameters, FLOPs and memory cost on ResNet-50 and ResNet-101 by training on ImageNet-1K. $^\dagger$ denotes training with the implemental settings in BA-Net.

| Model | Params | FLOPs | Input | Top-1 | Top-5 |
|---|---|---|---|---|---|
| BA-Net-50 (Zhao et al., 2022) | 28.7 M | 4.2 B | 224 | 77.8 | 93.7 |
| R50 + MRLA-light (Ours) | 25.7 M | 4.2 B | 224 | 77.7 | 93.8 |
| BA-Net-50$^\dagger$ (Zhao et al., 2022) | 28.7 M | 4.2 B | 224 | 78.9 | 94.3 |
| R50 + MRLA-light$^\dagger$ (Ours) | 25.7 M | 4.2 B | 224 | 78.7 | 94.4 |

### B.2 OUT-OF-MEMORY PROBLEM AND THE NECESSITY OF MRLA-LIGHT

Though it is possible to train a naive quadratic version with MLA (Eq. (4)) and MRLA-base (Eq. (6)) on ResNet-50 and obtain good results, it will lead to the out-of-memory (OOM) problem on ResNet-101 if we keep the same batch size of 256. Here, we compare the parameters, FLOPs and memory cost per GPU (all models are trained on 4 V100 GPUs) of MLA, MRLA-base and MRLA-light to support our claim.

Table 7: Comparisons of parameters, FLOPs and memory cost on ResNet-50 and ResNet-101 by training on ImageNet-1K.

| Model | Params | FLOPs | Memory (MiB) |
|---|---|---|---|
| ResNet-50 with MLA | 25.7 M | 4.8 B | 18.2 K |
| ResNet-50 with MRLA-base | 25.7 M | 4.6 B | 13.8 K |
| ResNet-50 with MRLA-light | 25.7 M | 4.2 B | 12.2 K |
| ResNet-101 with MLA | 44.9 M | 9.1 B | OOM |
| ResNet-101 with MRLA-base | 44.9 M | 8.5 B | OOM |
| ResNet-101 with MRLA-light | 44.9 M | 7.9 B | 17.7 K |

Table 7 shows that MLA and MRLA-base demand more extra FLOPs and memory than MRLA-light when the network becomes deeper. Moreover, ResNet-50 with MLA and MRLA-base cost 30% and 20% more time during the training period. Since ResNet-101 has 23 building blocks in the third stage, the layer attention for the last layer should attend to the features stacking all previous 22 layers' features. Unless we manually split them into several sub-stages for a deep network, MLA and MRLA-base for the last layer in the stage should attend to the features stacking all previous layers' features, leading to the OOM problem. In general, stacking features layer by layer is unfriendly to a deep network as it incurs significant computation, memory and time costs.

### B.3 OBJECT DETECTION AND INSTANCE SEGMENTATION ON COCO

**Implementation details** We adopt the commonly used settings (Hu et al., 2018; Wang et al., 2018; Cao et al., 2019; Wang et al., 2020b; Zhao et al., 2021), which are the default settings in MMDetection

Table 8: Complete results of Mask R-CNN on object detection using different methods. The bold fonts denote the best performances.

| Methods | Params | GFLOPs | $AP^{bb}$ | $AP^{bb}_{50}$ | $AP^{bb}_{75}$ | $AP^{bb}_S$ | $AP^{bb}_M$ | $AP^{bb}_L$ |
|---|---|---|---|---|---|---|---|---|
| ResNet-50 | 44.18 M | 275.58 | 37.2 | 58.9 | 40.3 | 22.2 | 40.7 | 48.0 |
| + SE (Hu et al., 2018) | 46.67 M | 275.69 | 38.7 | 60.9 | 42.1 | 23.4 | 42.7 | 50.0 |
| + ECA (Wang et al., 2020b) | 44.18 M | 275.69 | 39.0 | 61.3 | 42.1 | 24.2 | 42.8 | 49.9 |
| + 1 NL (Wang et al., 2018) | 46.50 M | 288.70 | 38.0 | 59.8 | 41.0 | - | - | - |
| + GC (r16) (Cao et al., 2019) | 46.90 M | 279.60 | 39.4 | 61.6 | 42.4 | - | - | - |
| + GC (r4) (Cao et al., 2019) | 54.40 M | 279.60 | 39.9 | 62.2 | 42.9 | - | - | - |
| + RLA$_g$ (Zhao et al., 2021) | 44.43 M | 283.06 | 39.5 | 60.1 | 43.4 | - | - | - |
| + BA (Zhao et al., 2022) | 47.3 0M | 261.98 | 40.5 | 61.7 | 44.2 | 24.5 | 44.3 | 52.1 |
| + MRLA-light (Ours) | 44.34 M | 276.93 | **41.2** | **62.3** | **45.1** | **24.8** | **44.6** | **53.5** |
| ResNet-101 | 63.17 M | 351.65 | 39.4 | 60.9 | 43.3 | 23.0 | 43.7 | 51.4 |
| + SE (Hu et al., 2018) | 67.89 M | 351.84 | 40.7 | 62.5 | 44.3 | 23.9 | 45.2 | 52.8 |
| + ECA (Wang et al., 2020b) | 63.17 M | 351.83 | 41.3 | 63.1 | 44.8 | 25.1 | 45.8 | 52.9 |
| + 1 NL (Wang et al., 2018) | 65.49 M | 364.77 | 40.8 | 63.1 | 44.5 | - | - | - |
| + GC (r16) (Cao et al., 2019) | 68.10 M | 354.30 | 41.1 | 63.6 | 45.0 | - | - | - |
| + GC (r4) (Cao et al., 2019) | 82.20 M | 354.30 | 41.7 | 63.7 | 45.5 | - | - | - |
| + RLA$_g$ (Zhao et al., 2021) | 63.56 M | 362.55 | 41.8 | 62.3 | 46.2 | - | - | - |
| + MRLA-light (Ours) | 63.54 M | 353.84 | **42.8** | **63.6** | **46.5** | **25.5** | **46.7** | **55.2** |

toolkit (Chen et al., 2019). [2] Specifically, the shorter side of input images are resized to 800, then all detectors are optimized using SGD with weight decay of 1e-4, momentum of 0.9 and batch size of 16. The learning rate is initialized to 0.02 and is decreased by a factor of 10 after 8 and 11 epochs, respectively, i.e., the 1x training schedule (12 epochs). For RetinaNet, we modify the initial learning rate to 0.01 to avoid training problems. Since the models no longer suffer from overfitting in these transfer learning tasks, we remove stochastic depth on our MRLA used in ImageNet classification.

**Results**   Complete results of Mask R-CNN on object detection and instance segmentation are shown in Tables 8 and 9 for comprehensive comparisons. We observe that our method brings clear improvements over the original ResNet on all the evaluation metrics. Compared with the two channel attention methods SE and ECA, our MRLA achieves more gains for small objects, which are usually more difficult to be detected. Interestingly, even though BA-Net utilizes the better pre-trained weight (the model with $^\dagger$ superscript in Table 6) in object detection and instance segmentation, our approach still outperforms the BA-Net counterpart on most of the metrics.

Besides the comparisons with various attention methods using the same baseline models, we also compare our model with different types of networks, as shown in Table 10.

### B.4   DISCUSSION ON ABLATION STUDY

**MLA and MRLA-base are both effective.**   Before introducing MRLA-light, we also proposed the MLA (referring to multi-head version of Eq. (4) in Section 3.2) and the MRLA-base (referring to Eq. (9) in Section 3.4), both of which rigorously follow the definition of self-attention in Transformer (Vaswani et al., 2017). From Tables 1 and 4, we can observe that MLA and MRLA-base perform better than most of other methods. Besides, there is only a negligible gap between their performance and that of MRLA-light in some cases, which can be attributed to the following reasons:

- We directly applied MRLA-light's hyper-parameter setting and design to the other two without further tuning.
- Benefiting from simpler architecture and learnable $\lambda_o^t$, MRLA-light is easier to train and more flexible.

Therefore, if we do not chase the minimum computation, time and memory cost, MLA and MRLA-base are also good choices as they can equivalently enrich the representation power of a network as MRLA-light.

---

[2]License: Apache License 2.0

Table 9: Complete results of Mask R-CNN on instance segmentation using different methods. The bold fonts denote the best performances.

| Methods | Params | GFLOPs | $AP^m$ | $AP^m_{50}$ | $AP^m_{75}$ | $AP^m_S$ | $AP^m_M$ | $AP^m_L$ |
|---|---|---|---|---|---|---|---|---|
| ResNet-50 | 44.18 M | 275.58 | 34.1 | 55.5 | 36.2 | 16.1 | 36.7 | 50.0 |
| + SE (Hu et al., 2018) | 46.67 M | 275.69 | 35.4 | 57.4 | 37.8 | 17.1 | 38.6 | 51.8 |
| + ECA (Wang et al., 2020b) | 44.18 M | 275.69 | 35.6 | 58.1 | 37.7 | 17.6 | 39.0 | 51.8 |
| + 1 NL (Wang et al., 2018) | 46.50 M | 288.70 | 34.7 | 56.7 | 36.6 | - | - | - |
| + GC (r16) (Cao et al., 2019) | 46.90 M | 279.60 | 35.7 | 58.4 | 37.6 | - | - | - |
| + GC (r4) (Cao et al., 2019) | 54.40 M | 279.60 | 36.2 | 58.7 | 38.3 | - | - | - |
| + RLA$_g$ (Zhao et al., 2021) | 44.43 M | 283.06 | 35.6 | 56.9 | 38.0 | - | - | - |
| + BA Zhao et al. (2022) | - | - | 36.6 | 58.7 | 38.6 | 18.2 | 39.6 | **52.3** |
| + MRLA-light (Ours) | 44.34 M | 276.93 | **37.1** | **59.1** | **39.6** | **19.5** | **40.3** | 52.0 |
| ResNet-101 | 63.17 M | 351.65 | 35.9 | 57.7 | 38.4 | 16.8 | 39.1 | 53.6 |
| + SE (Hu et al., 2018) | 67.89 M | 351.84 | 36.8 | 59.3 | 39.2 | 17.2 | 40.3 | 53.6 |
| + ECA (Wang et al., 2020b) | 63.17 M | 351.83 | 37.4 | 59.9 | 39.8 | 18.8 | 41.1 | 54.1 |
| + 1 NL (Wang et al., 2018) | 65.49 M | 364.77 | 37.1 | 59.9 | 39.2 | - | - | - |
| + GC (r16) (Cao et al., 2019) | 68.10 M | 354.30 | 37.4 | 60.1 | 39.6 | - | - | - |
| + GC (r4) (Cao et al., 2019) | 82.20 M | 354.30 | 37.6 | 60.5 | 39.8 | - | - | - |
| + RLA$_g$ (Zhao et al., 2021) | 63.56 M | 362.55 | 37.3 | 59.2 | 40.1 | - | - | - |
| + BA (Zhao et al., 2022) | - | - | 38.1 | **60.6** | 40.4 | 18.7 | 41.5 | **54.8** |
| + MRLA-light (Ours) | 63.54 M | 353.84 | **38.4** | **60.6** | **41.0** | **20.4** | **41.7** | **54.8** |

Table 10: Object detection results with different backbones using RetinaNet as a framework on COCO val2017. All models are trained in "1x" schedule. FLOPs are calculated on $1280 \times 800$ input. The blue bold fonts denote the best performances, while the bold ones perform comparably.

| Backbone Model | Params | GFLOPs | $AP^{bb}$ | $AP^{bb}_{50}$ | $AP^{bb}_{75}$ | $AP^{bb}_S$ | $AP^{bb}_M$ | $AP^{bb}_L$ |
|---|---|---|---|---|---|---|---|---|
| ResNet-101 (He et al., 2016) | 56.7 M | 315 | 37.7 | 57.5 | 40.4 | 21.1 | 42.2 | 49.5 |
| RelationNet++ (Chi et al., 2020) | 39.0 M | 266 | 39.4 | 58.2 | 42.5 | - | - | - |
| ResNeXt-101-32x4d (Xie et al., 2017) | 56.4 M | 319 | 39.9 | 59.6 | 42.7 | 22.3 | 44.2 | 52.5 |
| Swin-T (Liu et al., 2021) | 38.5 M | 245 | **41.5** | **62.1** | **44.2** | **25.1** | 44.9 | **55.5** |
| MRLA-ResNet-101 (Ours) | 57.1 M | 318 | **41.3** | 61.4 | **44.2** | **24.8** | **45.6** | 53.8 |

**Convolutions in Transformers**   The improvement of our MRLA over transformers is not entirely caused by the convolution and the experiments in Table 11 support this point:

- We have inserted a DWConv layer into DeiT which is a convolution-free transformer. The result demonstrates that our MRLA outperforms adding the DWConv layer.

- We have also applied our MRLA to some convolutional transformers, e.g., CeiT (Yuan et al., 2021) and PVTv2 (Wang et al., 2022a). We can find that our MRLA can further boost the performances of these convolutional transformers.

**Stochastic Depth**   Stochastic depth is not the fundamental component that helps MRLA outperform its counterparts. Instead, it is a tool to avoid overfitting (too-high training accuracy) on the middle-size dataset ImageNet-1K. We have found that MLA and MRLA-base also suffer from overfitting problem though they are less severe than that in MRLA-light. As we stated in Section 1, strengthening layer interactions can improve model performances. We then conjecture that the information from previous layers brought by our MRLAs is too strong, leading to the overfitting. The detailed justifications are as follows:

- Applying stochastic depth with the same survival probability on the "+ DWConv2d" and ECA module does not bring significant improvements (see Table 12 (a) and (c)), proving that stochastic depth itself is not the key to boosting the model performance.

- It is natural to share the same stochastic depth on EfficientNet and vision transformers since the layer attention should not be applied if that layer is dropped.

Table 11: Performances of different transformers with our MRLA-light and DeiT-T with additional convolutions.

| Model | Top-1 | Top-5 |
|---|---|---|
| DeiT-T (Touvron et al., 2021) | 72.2 | 91.1 |
| + DWConv | 72.8 | 91.7 |
| + MRLA-light (Ours) | 73.4 | 91.9 |
| CeiT-T (Yuan et al., 2021) | 76.4 | 93.4 |
| + MRLA-light (Ours) | 77.4 | 94.1 |
| CeiT-S (Yuan et al., 2021) | 82.0 | 95.9 |
| + MRLA-light (Ours) | 83.2 | 96.6 |
| PVTv2-B0 (Wang et al., 2022a) | 70.5 | - |
| + MRLA-light (Ours) | 71.5 | 90.7 |
| PVTv2-B1 (Wang et al., 2022a) | 78.7 | - |
| + MRLA-light (Ours) | 79.4 | 94.9 |

- For object detection and instance segmentation on COCO, we did not observe any overfitting problem when removing the stochastic depth trick. We speculate that the 12-epoch training leads to underfitting since we adopt the standard 1x training schedule. Therefore, there is no need to use this trick for these two tasks.

- There are indeed other solutions to address the overfitting problem but they are sub-optimal.

  – We prefer using stochastic depth over pretraining on larger datasets because of limited computational resources and time. The ImageNet-22K (14M) and JFT300M (300M) datasets are significantly larger than the ImageNet-1K (1.28M). Besides, choosing this more efficient strategy allows a fair comparison with current SOTA attention models, as most of them are not pretrained on these larger datasets.

  – Mixup augmentation (M) and label smoothing (LS) were also tried to prevent overfitting. Using them simultaneously can achieve similar performance to the stochastic depth (see Table 12 (b)). However, these methods influence the entire network instead of our MRLA only, leading to unfair comparisons with other models.

  – Manually applying MRLA to partial layers instead of stochastic depth is also feasible. However, it maycost much more time to decide MRLA at which layer should be dropped.

Table 12: Ablation study on the trick of stochastic depth.

| Model | Params | FLOPs | Top-1 |
|---|---|---|---|
| ResNet-50 | 25.6 M | 4.1 B | 76.1 |
| (a) + DWConv2d | 25.7 M | 4.2 B | 76.6 |
|    - w/ stochastic | 25.7 M | 4.2 B | 76.9 |
| (b) MRLA-light (M,LS) | 25.7 M | 4.2 B | 77.9 |
| (c) R50 + ECA | 25.6 M | 4.1 B | 77.5 |
|    - w/ stochastic | 25.6 M | 4.1 B | 77.5 |

## B.5 VISUALIZATIONS

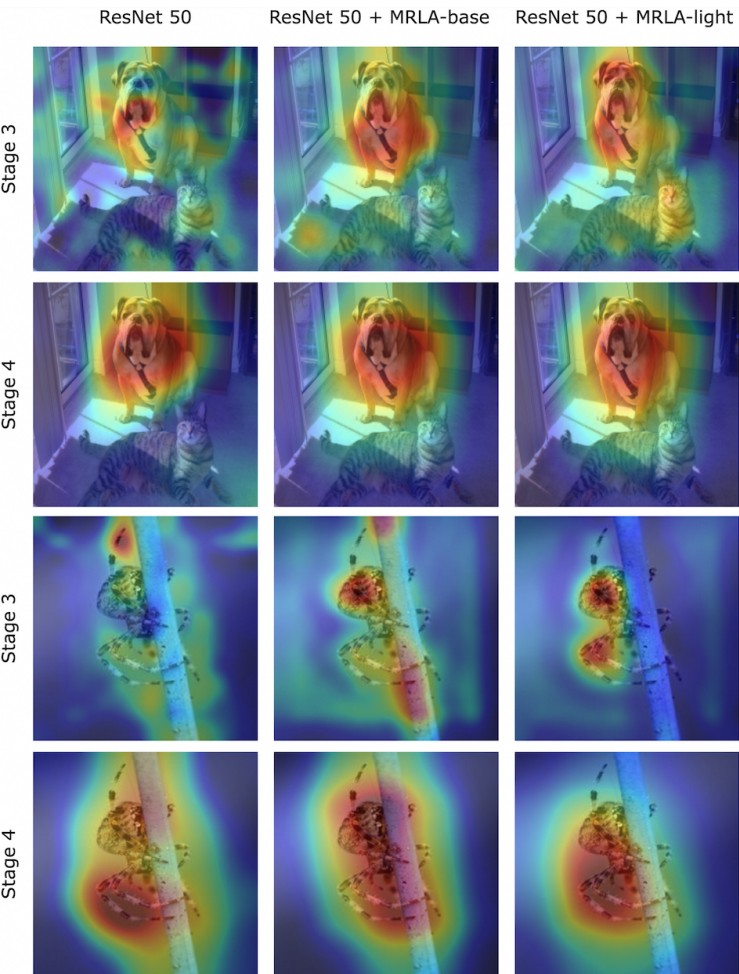

Figure 6: Visualizations of the feature maps extracted from the end of Stage 3 and 4 of ResNet-50 and our MRLA counterparts.

To investigate how MRLAs contribute to the representation learning in CNNs, we visualize the feature maps with the score-weighted visual explanations yielded by ScoreCAM (Wang et al., 2020a) in Figure 6. Specifically, we extract the feature maps from the end of each stage in ResNet-50 and our MRLA counterparts. We display here the visualizations for Stage 3 and 4 as the feature maps from the first two stages are quite similar and focus on low-level features for all models. The two example images are randomly selected from the ImageNet validation set. In the visualizations, the area with the warmer color contributes more to the classification. We can observe that: (1) The models with MRLAs tend to find the critical areas faster than the baseline model. Especially in stage 3, the MRLAs have already moved to emphasize the high-level features while the baseline model still focuses on the lower-level ones. (2)The areas with the red color in ResNet 50 + MRLA-base/light models are larger than that in the baseline model, implying that the MRLA counterparts utilize more information for the final decision-making. (3) The patterns of MRLA-base and MRLA-light are similar, validating that our approximation in MRLA-light does not sacrifice too much of its ability.

Figure 7 visualizes the attention maps of a specified query (red box) from three randomly chosen heads in the last layer of DeiT-T and our MRLA counterparts. The first image is randomly sampled from the ImageNet validation set, and the second image is downloaded from a website. [3] In the

---

[3]https://github.com/luo3300612/Visualizer

visualization, the area with the warmer color has a higher attention score. We can observe that MRLA can help the network retrieve more task-related local details compared to the baseline model. In other words, the low-level features are better preserved with layer attention.

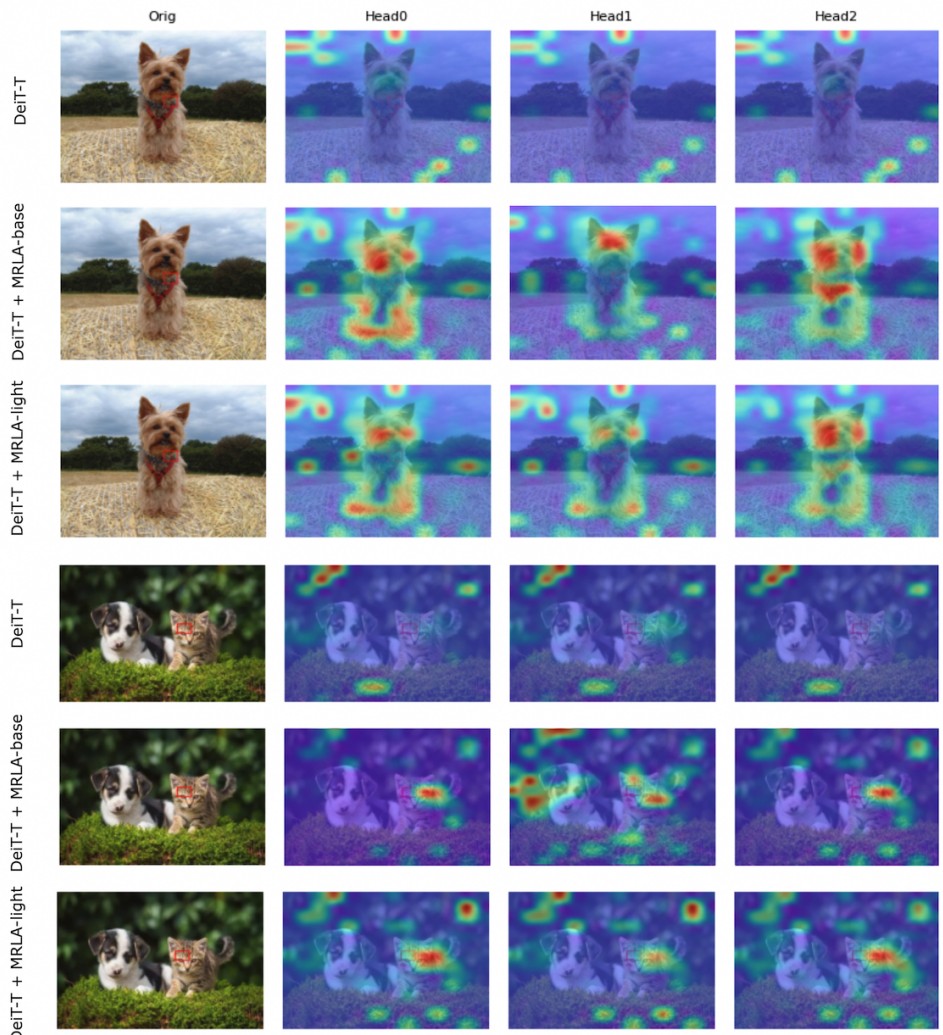

Figure 7: Visualizations of the attention maps in the last layer of DeiT-T and our MRLA counterpart given a query specified in the red box.

