# OpenReview forum: "Cross-Layer Retrospective Retrieving via Layer Attention"
_ICLR.cc/2023/Conference — ICLR 2023 poster_

### Official Review · Reviewer_Gvtx · 2022-10-22

**Confidence:** 3
**Correctness:** 3
**Technical Novelty And Significance:** 3
**Empirical Novelty And Significance:** 3
**Recommendation:** 5

**Clarity, Quality, Novelty And Reproducibility:**

The paper is generally written well and easy to read. The formulation of the proposed layer attention is similar to that of token attention in transformers. The code is provided for reproduction.

**Strength And Weaknesses:**

Strong points.

1. The paper proposes a new attention mechanism by formulating each layer as a token and querying all previous layers.
2. The paper formulates extends the proposed layer attention to have multi-head and be light-weighted.
3. Code is submitted.


Weak points.

1. It is not clear to me why Eq.5 can be derived from Eq.3. More explanation would be better. Besides, it seems there are some symbol ambiguities, for example, it is not clear what the differences are between f and F. in Eq.3 and Eq.5, and what is s in Eq. 1 and Eq. 4.

2. Section 3.3 mentions an assumption that query vectors at two consecutive layers have a similar pattern, based on which the linearization is performed by using a learnable parameter. However, such an assumption is not justified. The paper should also compare the method proposed by (Katharopoulos et al. 2020) to demonstrate its own advantages.

3. Almost all papers referred to are published before/in 2021. A more comprehensive literature review and experimental comparisons are essential in order to evaluate its novelties.

**Summary Of The Paper:**

This paper proposes a multi-head recurrent layer attention (MRLA) mechanism, which uses the feature of the current layer to query all previous layers to retrieve query-related information of different levels. The paper further proposes a light-weighted MRLA to reduce the computational cost. The proposed method can be applied to both CNN and transformer-based networks. Experiments show its effectiveness in several vision tasks including image classification, object detection, and instance segmentation.

**Summary Of The Review:**

I am on the negative side at this point mainly due to the out-of-date literature review, and some experiments that support key assumptions/claims are missing.

---

> ### Author Response · Authors · 2022-11-16
> **To Reviewer Gvtx (Part III)**
>
> **3. Response to W3: Updated literature review**
>
> Thanks for your careful reading. After a thorough search of the latest literature, we have updated our Introduction and Related Work (Sec. 1 and 2). Please refer to the highlighted parts in the revised draft for details.
>
> Specifically, we have found the following five papers:
>
> * CANet (Li et al., 2021), which is specially designed for small object detection in the field of remote sensing. It integrates the features of different layers by resizing and averaging the feature maps. There is no cross-layer attention involved and its attention module is similar to the NL block and operates within a layer.
> * TDAM (Jaiswal et al., 2022) and BANet (Zhao et al., 2022), both of which perform attention on low- and high-level feature maps within a convolutional bottleneck block. Note that in our layer attention design, we consider a convolutional bottleneck block as a layer and thus TDAM and BANet are considered as inner-layer attention under the concept. Therefore, our layer attention that strengthens interactions among convolutional bottleneck blocks can be added on top of their modules.
> * ACLA (Wang et al., 2022b), which extends the NL block (Wang et al., 2018) and is very close to the OmniNet (Tay et al., 2021). ACLA allows each spatial token from each layer to attend to all spatial tokens from the previous layers and itself. Importantly, it neglects these tokens' layer identity, which, in contrast, our MRLA emphasizes. To avoid the high computation cost, ACLA samples key tokens from each layer and adds hard gating masks, leading to an attention with incomplete information. Notably, ACLA is designed for image restoration and it is validated on the datasets and tasks that are non-overlapped with our paper.
> * CN-CNN (Guo et al., 2022), which first uses DIANet's LSTM to aggregate features and then performs the spatial and channel attention. It is validated on fine-grained visual classification datasets.
>
> References:
>
> 1. Yangyang Li, Qin Huang, Xuan Pei, Yanqiao Chen, Licheng Jiao, and Ronghua Shang. Cross-layer attention network for small object detection in remote sensing imagery. *IEEE Journal of Selected Topics in Applied Earth Observations and Remote Sensing*, 14:2148–2161, 2021. doi: 10.1109/JSTARS.2020.3046482.
> 2. Shantanu Jaiswal, Basura Fernando, and Cheston Tan. TDAM: Top-down attention module for contextually guided feature selection in cnns. In  *European Conference on Computer Vision*, pp. 259–276, 2022.
> 3. Yue Zhao, Junzhou Chen, Zirui Zhang, and Ronghui Zhang. BA-Net: Bridge attention for deep convolutional neural networks. In *European Conference on Computer Vision*, pp. 297–312, 2022.
> 4. Yancheng Wang, Ning Xu, Chong Chen, and Yingzhen Yang. Adaptive cross-layer attention for image restoration. CoRR, abs/2203.03619, 2022b. doi: 10.48550/arXiv.2203.03619. URL https://doi.org/10.48550/arXiv.2203.03619.
> 5. Xiaolong Wang, Ross Girshick, Abhinav Gupta, and Kaiming He. Non-local neural networks. In *Proceedings of the IEEE conference on computer vision and pattern recognition*, pp. 7794–7803,2018.
> 6. Yi Tay, Mostafa Dehghani, Vamsi Aribandi, Jai Gupta, Philip M Pham, Zhen Qin, Dara Bahri, Da-Cheng Juan, and Donald Metzler. Omninet: Omnidirectional representations from transformers. In Marina Meila and Tong Zhang (eds.), *Proceedings of the 38th International Conference on*
>    *Machine Learning*, volume 139 of Proceedings of Machine Learning Research, pp. 10193–10202. PMLR, 18–24 Jul 2021.
> 7. Chenyu Guo, Jiyang Xie, Kongming Liang, Xian Sun, and Zhanyu Ma. Cross-layer navigation convolutional neural network for fine-grained visual classification. In *ACM Multimedia Asia*, MMAsia ’21, 2022.

---

> ### Author Response · Authors · 2022-11-16
> **To Reviewer Gvtx (Part II)**
>
> **2. Response to W2**
>
> **(1) Assumption about the query vectors at two consecutive layers have a similar pattern**
>
> Thanks for your constructive comment. Since the multi-head design is adopted, we validate whether $\boldsymbol{Q}^t_h$ is roughly proportional to $\boldsymbol{Q}^{t-1}_h$ following your suggestion. Please see the resulting histogram in Figure 2(b) and the description at the end of Sec. 5.1.
>
> Specifically, we compute the absolute value of the cosine similarity between the queries that are from consecutive layer attention blocks of the same stage, which is equal to $|\frac{\boldsymbol{Q}^t\_h \cdot \boldsymbol{Q}^{t-1}\_h}{||\boldsymbol{Q}^t\_h|| ||\boldsymbol{Q}^{t-1}\_h||}|.$ Note that if the value approaches 1, we will achieve the desire that the elements of $\boldsymbol{\lambda}_{q,h}^t$ have similar values.
>
> We conduct the validation by randomly sampling 5 images from each class of the ImageNet validation dataset and the total number of images is 5000. We classify these images with the trained ResNet-50 + MRLA-base model. And then we extract the query vectors from each head of all MRLA-base blocks except for those belonging to the first layer within each stage. The exclusion is due to the fact that the first layer of each stage only attends to itself but not to other layers.
>
> **(2) Comparison with linear attention technique proposed by (Katharopoulos et al., 2020)**
>
> We are sorry for not being clear in our previous draft and we have updated our Sec 3.2 to elaborate the role of other linearization techniques to our layer attention.
>
> First, we would like to emphasize that the main contribution of our paper is orthogonal to that of linear transformers proposed by Katharopoulos et al., 2020. The linear transformers linearize the complexity of the autoregressive self-attention for the tokens within a layer, while our paper devises the layer attention that aims to strengthen layer interactions.
>
> Specifically, we first propose the formulation of layer attention (Eq.4) and its recurrent version MRLA-base (Eq.6). Then to adapt for deeper networks given the time and memory constraints, we also devise a light-weighted version MRLA-light (Eq.8).
>
> Therefore, any existing methods to linearize the complexity of autoregressive self-attention among tokens are complements to our recurrent layer attention if they work well at the layer level. We believe that the linearization technique is a tool that facilitates the implementation of layer attention under limited resources. This is also the reason why we provided the formulation of the linear recurrent layer attention with Katharopoulos et al.'s method in A.3 of the Appendix.
>
> Nonetheless, we have compared our MRLA-light with the version of Katharopoulos et al.'s linearization empirically. Their single-crop accuracies on the ImageNet-1K validation set are as follow:
>
> | Model                                                 | Top-1    | Top-5    |
> | :---------------------------------------------------- | :------- | :------- |
> | MRLA-light                                            | **77.7** | **93.8** |
> | MRLA with Katharopoulos et al.'s linearization method | 77.4     | 93.6     |

---

> ### Author Response · Authors · 2022-11-16
> **To Reviewer Gvtx (Part I)**
>
> Thanks very much for your valuable suggestions and comments. We sincerely appreciate your time in reading the paper, and our point-to-point responses to your comments are given below.
>
> **1. Response to W1**
>
> **(1) The derivation from Eq.3 to Eq.5**
>
> We apologize for missing some details in the main paper. In the vanilla self-attention, suppose we have $\boldsymbol{X} \in \mathbb{R}^{{T} \times D_{in}}$ where ${T}$ stands for the number of tokens and ${D_{in}}$ is the input feature dimension. The vanilla self-attention derives the key and value matrices $\boldsymbol{K} \in \mathbb{R}^{T \times D_k}$ and $\boldsymbol{V} \in \mathbb{R}^{T \times D_{out}}$ by applying the transformation functions $F_K$ and $F_V$ respectively:
> $$
> \boldsymbol{K} = F\_K(\boldsymbol{X}) = F\_K(\text{Concat}[\boldsymbol{X}\_{1,:}, \boldsymbol{X}\_{2,:}, ..., \boldsymbol{X}\_{T,:}])
> \quad\text{and}\quad\boldsymbol{V} = F\_V(\boldsymbol{X})=F\_V(\text{Concat}[\boldsymbol{X}\_{1,:}, \boldsymbol{X}\_{2,:}, ..., \boldsymbol{X}\_{T,:}]).$$
>
> Importantly, both the transformation functions are linear projections, i.e., $F_K(\boldsymbol{X}) = XW_K$ and $F_V(\boldsymbol{X})=XW_V$. They only operate on the feature dimension $D_{in}$ and thus
> $$\boldsymbol{K}=\text{Concat} [F\_K(\boldsymbol{X}\_{1,:}), ..., F_K(\boldsymbol{X}\_{{T},:})] \quad \text{and}
> \quad \boldsymbol{V}=\text{Concat}[F\_V(\boldsymbol{X}\_{1,:}), ..., F_V(\boldsymbol{X}\_{T,:})]. $$
>
>    Recall that in layer attention, we assume $D_{in} = D_{out} = D$ and denote the output feature of the $t$-th layer as $\boldsymbol{X}^t \in \mathbb{R}^{1\times D}$. In the $t$-th layer attention, we derive the key and value $\boldsymbol{K}^t \in \mathbb{R}^{t\times D_k}$ and $\boldsymbol{V}^t \in \mathbb{R}^{t\times D}$ by applying the transformation functions $f_K^t$ and $f_V^t$ as in Eq.3:
>
>    $$
>    \begin{align*}
>    	\boldsymbol{K}^t = \text{Concat}[f_K^t(\boldsymbol{X}^1),...,f_K^t(\boldsymbol{X}^t)] \quad \text{and} \quad \boldsymbol{V}^t = \text{Concat}[f_V^t(\boldsymbol{X}^1),...,f_V^t(\boldsymbol{X}^t)],
>    \end{align*}
>    $$
>
>    which directly borrows the formulation from vanilla attention.
>    Compared with the Eq.5:
>
>    $$
>    \begin{equation*}
>      \boldsymbol{K}^t = \text{Concat}[f_K^1(\boldsymbol{X}^1),...,f_K^t(\boldsymbol{X}^t)] \hspace{3mm}\text{and}\hspace{3mm}
>      \boldsymbol{V}^t = \text{Concat}[f_V^1(\boldsymbol{X}^1),...,f_V^t(\boldsymbol{X}^t)],
>    \end{equation*}
>    $$
>
>    Eq. 3 has larger computation complexity. This is because, in Eq.3, $\boldsymbol{K}^{t-1} = \text{Concat}[f_K^{t-1}(\boldsymbol{X}^1),...,f_K^{t-1}(\boldsymbol{X}^{t-1})]$ and thus $\boldsymbol{K}^t$ cannot inherit anything from $\boldsymbol{K}^{t-1}$. It is then natural to use the same transformation functions to avoid redundancy that is caused by repeatedly deriving the keys and values for the same layer with different transformation functions.
>
> Due to the limited space in the main paper, we included the explanation about the simplification in Eq.5 in A.2 of our appendix in the initial submission.
>
> **(2) The meaning of notation $s$ in Eq.1 and Eq.4**
>
> In both Eq.1 and Eq.4, $s$ stands for the index that is iterated in the summation. Here we adopt NumPy-like notations, where for a matrix $\boldsymbol{Y} \in \mathbb{R}^{I \times J}$, $\boldsymbol{Y}\_{i,:}$ is its $i$-th row. Hence in Eq.1, $\boldsymbol{V}\_{s,:} \in \mathbb{R}^{1\times D\_{out}}$ refers to the $s$-th row of the value matrix $\boldsymbol{V} \in \mathbb{R}^{T \times D_{out}}$, which is the value representation of the $s$-th token. Similarly in Eq.4, $\boldsymbol{V}^t_{s,:} \in \mathbb{R}^{1\times D}$ refers to the $s$-th row of the $t$-th layer attention's value matrix $\boldsymbol{V}^t \in \mathbb{R}^{t \times D}$.
>
> To make the symbols more consistent, we have revised the notations in Sec 3.1. As Sec 3.1 revisits the vanilla self-attention that does not have any mask, the $t$-th query token can attend to all $T$ tokens. In contrast, in the later subsections, since layer attention is causal, the $t$-th layer can only attend to all previous layers and itself. Hence $\boldsymbol{V} \in \mathbb{R}^{T \times D_{out}}$ in Eq.1 while $\boldsymbol{V}^t \in \mathbb{R}^{t \times D}$ in Eq.4.
>
> **(3) The notations of $F(\cdot)$ and $f(\cdot)$ in Eq.3 and Eq.5**
>
> Thank you for the careful reading. Previously, we used $F(\cdot)$ and $f(\cdot)$ to show that the learning parameters are different. However, to avoid ambiguity, we have revised the notation following your suggestion.

---

> ### Author Response · Authors · 2022-11-24
> **Summary of our responses**
>
> Thank you again for all the insightful comments. According to them, we have revised our draft and believe that the quality of our paper has been further improved. Here is a summary of our revisions:
>
> 1. We have included a detailed explanation of the derivation from Eq.3 to Eq.5 in Appendix A.2.
> 2. We have revised the notations for consistency in Sec 3.1.
> 3. We have changed the notation of the feature extraction function from $F(\cdot)$ to $f(\cdot)$ in Sec 3.2 to avoid ambiguity.
> 4. We have verified the assumption that query vectors at two consecutive layer attention blocks have a similar pattern, and added the visualization in Figure 2(b) and the detailed explanation in Sec 5.1.
> 5. We have elaborated on the relationship between our layer attention and other attention linearization techniques in Sec 3.2.
> 6. We have conducted a more comprehensive literature review and added the recently published works in Sec 1 and 2.
>
> The details are included in our previous point-to-point responses. We sincerely hope they can resolve your concerns and welcome further discussion if you have any questions.

---

> ### Author Response · Authors · 2022-12-05
> **We'd love to know if you have any more questions after our response**
>
> Dear Reviewer Gvtx,
>
> Thank you very much for your insightful comments and suggestions, which are very helpful for us to improve the paper's quality. We have tried our best to carefully address all of your comments in our response and the revised manuscript. Please kindly let us know if you have any further questions, and we are delighted to follow up.
>
> Thank you for your time and attention.
>
> Best regards,
>
> ICLR 2023 Conference Paper6346 Authors

---

> ### Author Response · Authors · 2022-12-07
> **We would be really grateful if you could kindly let us know whether our responses have addressed your concerns.**
>
> Dear Reviewer Gvtx,
>
> As we are approaching the end of the rebuttal period, we would be really grateful if you could kindly let us know whether our responses have addressed your concerns. If you have any further questions, we are more than happy to follow up.
>
> Thank you for your time and attention.
>
> Best regards,
>
> ICLR 2023 Conference Paper6346 Authors

---

> ### Author Response · Authors · 2022-12-12
> **We are eager for your feedbacks**
>
> Dear Reviewer Gvtx,
>
> It comes to the end of the rebuttal period. Could you please give us a simple comment on our revisions and responses even if you may not be satisfied with them? This is very important for us, say, for another submission if we fail this conference. Many thanks in advance.
>
> Best regards,
>
> ICLR 2023 Conference Paper6346 Authors

---

> > ### Comment · Reviewer_Gvtx · 2022-12-12
> > **Acknowledgement of responses**
> >
> > Dear authors,
> >
> > Thank you for the responses and sorry for the late reply.
> >
> > As indicated before, I am NOT an expert in this specific area. But honestly, I feel that the idea of studying cross-layer dependencies with attention in a recursive manner is incremental, so I expect the merits to be the performance and analysis. I would like to suggest the authors consider the following two points.
> >
> > 1. I briefly take a look at the TDAM (Jaiswal et al., 2022), BANet (Zhao et al. , 2022), and ACLA (Wang et al., 2022b) papers. In my opinion, all these three papers study the exact same cross-layer attention but in different ways of implementation compared to the proposed method. I understand that ACLA is currently an arxiv paper, and TDAM and BANet are recently published in ECCV'22. Nonetheless, I think it would be better if authors could quantitatively compare the proposed method to these existing methods.
> >
> > 2. The performance gain seems not significant but I am not sure whether the performance is already saturated. If so, the authors may consider applying the proposed method to other vision tasks.
> >
> > I can be neutral but I do not find a strong reason to push me to raise my rating. Sorry for this.
> >
> > Best wishes,
> > Gvtx

---

> > > ### Author Response · Authors · 2022-12-12
> > > **Response to Point 1 - Methodology**
> > >
> > > We really appreciate your feedback on our revisions and we would like to make some clarifications.
> > > We are afraid that we cannot agree that TDAM, BANet, and ACLA study the exact same cross-layer attention but in different ways of implementations compared to our proposed method. We wish to draw your attention to the fact that they are substantially different from our MRLA. The details are as below.
> > >
> > > **TDAM**
> > >
> > > * TDAM only operates within a standard convolutional block as they empirically find that having a large feedback-distance between the bottom representation $\boldsymbol{X_t^0}$ and the top representation $\boldsymbol{X_t^N}$ leads to **unstable training and significantly worsens the performance**. In contrast, our layer attention considers a convolutional block in CNNs as a layer and operates across multiple layers. **Consistent improvements** can be observed when applying to MRLAs to deep models, including ResNet-152 and DeiT-B.
> > >
> > > **BANet**
> > >
> > > * To **avoid a significant increase in complexity**, BANet also merely considers the features within the convolutional block, which is regarded as a layer in our layer attention. Our layer attention can therefore add on top of the BANet. Importantly, our MRLA-light **only has a linear computational complexity** with respect to the network depth, which can be easily applied to a very deep network.
> > >
> > > **ACLA**
> > >
> > > * ACLA is an extension of the NL block, which attends to all spatial locations from all layers. Same as the OmniNet that we compare in Introduction, **it does not distinguish which layer the token comes from.** Instead, our layer attention emphasizes the layer identity of the tokens.
> > > * To avoid the high computational complexity in computing attention weights, ACLA **generates the attention weights from the query feature alone**. It is inconsistent with our definition of layer attention weights as it should depict the relative importance of the previous layers (keys) to the current layer (query).
> > > * ACLA samples keys from each layer; while, in contrast, our layer attention utilizes all information from a layer.

---

> > > > ### Author Response · Authors · 2022-12-12
> > > > **Response to Point 1 - Empirical Comparison (Part I)**
> > > >
> > > > We agree with you that it's important to **quantitatively compare** our MRLA with other methods. As it is the last moment of the rebuttal period, we are afraid that we cannot provide you with additional experiment results. However, we can use some existing results in their papers to make rough comparisons.
> > > >
> > > > **TDAM**
> > > >
> > > > First, TDAM and our paper adopt different implementation versions of the baseline model. We use the results from the torchvision  toolkit while TDAM utilizes those from the pytorch-image-models (timm). **Note that the latter implementation includes advanced design settings and training tricks to improve the performance of ResNets.** **Therefore, it would be unfair to directly evaluate the performances between TDAM and MRLAs using the current results.** We will use a unified setting to reproduce the results of TDAM and our MRLA in the final version of our paper.
> > > >
> > > > Nonetheless, to shed light on the differences, we compare the increments in Top-1 and Top-5 accuracy on ImageNet-1K validation set, which are more reliable measures. From Tables 1 and 2, one can observe that our MRLA brings similar or even larger improvement overs the baseline models. Meanwhile, MRLA-light has much smaller numbers of parameters and FLOPs that the TDAM.
> > > >
> > > > Table 1: Comparisons of single-crop accuracy on the ImageNet-1K validation set provided by our paper.
> > > >
> > > > | Model                   | Params | FLOPs | Input | Top-1 | Top-5 | $\Delta$ Top-1 | $\Delta$ Top-5 |
> > > > | ----------------------- | ------ | ----- | ----- | ----- | ----- | ---------------- | ---------------- |
> > > > | ResNet-50 (Torchvision) | 25.6 M | 4.1 B | 224   | 76.1  | 92.9  | -                | -                |
> > > > | + SE                    | 28.1 M | 4.1 B | 224   | 76.7  | 93.4  | + 0.6            | + 0.5            |
> > > > | + CBAM                  | 28.1 M | 4.2 B | 224   | 77.3  | 93.7  | + 1.2            | + 0.8            |
> > > > | + ECA                   | 25.6 M | 4.1 B | 224   | 77.5  | 93.7  | + 1.4            | + 0.8            |
> > > > | + MRLA-base (Ours)      | 25.7 M | 4.6 B | 224   | 77.7  | 93.9  | + 1.6            | + 1.0            |
> > > > | + MRLA-light (Ours)     | 25.7 M | 4.2 B | 224   | 77.7  | 93.8  | + 1.6            | + 0.9            |
> > > > | + MRLA-light† (Ours)    | 25.7 M | 4.2 B | 224   | 78.2  | 94.1  | + 2.1            | + 1.2            |
> > > > | ResNet-101(Torchvision) | 44.5 M | 7.8 B | 224   | 77.4  | 93.5  | -                | -                |
> > > > | + SE                    | 49.3 M | 7.8 B | 224   | 77.6  | 93.9  | + 0.2            | + 0.4            |
> > > > | + CBAM                  | 49.3 M | 7.9 B | 224   | 78.5  | 94.3  | + 1.1            | + 0.8            |
> > > > | + MRLA-light (Ours)     | 44.9 M | 7.9 B | 224   | 78.7  | 94.4  | + 1.3            | + 0.9            |
> > > >
> > > > † Here we add a very light-weighted spatial scaling to our MRLA.
> > > >
> > > > Table 2: Comparisons of single-crop accuracy on the ImageNet-1K validation set provided by TDAM.
> > > >
> > > > | Method                   | Params | FLOPs | Input | Top-1 | Top-5 | $\Delta$ Top-1 | $\Delta$ Top-5 |
> > > > | ------------------------ | ------ | ----- | ----- | ----- | ----- | ---------------- | ---------------- |
> > > > | ResNet-50 (timm)         | 25.6 M | 4.1 B | 224   | 77.5  | 93.6  | -                | -                |
> > > > | + SE                     | 28.1 M | 4.1 B | 224   | 78.0  | 93.9  | + 0.5            | + 0.3            |
> > > > | + CBAM                   | 28.1 M | 4.2 B | 224   | 78.6  | 94.0  | + 1.1            | + 0.4            |
> > > > | + ECA                    | 25.6 M | 4.1 B | 224   | 78.1  | 93.9  | + 0.6            | + 0.3            |
> > > > | + TDjoint (t=2, m=1)     | 27.7 M | 4.6 B | 224   | 79.0  | 94.2  | + 1.5            | + 0.6            |
> > > > | + TDtop (t=2, m=1)       | 27.1 M | 4.6 B | 224   | 78.8  | 94.0  | + 1.3            | + 0.4            |
> > > > | + TDtop (t=2, m=3)       | 27.7 M | 6.0 B | 224   | 78.9  | 94.2  | + 1.4            | + 0.6            |
> > > > | ResNet-101 (timm)        | 44.5 M | 7.8 B | 224   | 80.4  | 95.3  | -                | -                |
> > > > | + SE                     | 49.3 M | 7.8 B | 224   | 80.8  | 95.4  | + 0.4            | + 0.1            |
> > > > | + CBAM                   | 49.3 M | 7.9 B | 224   | 81.2  | 95.6  | + 0.8            | + 0.3            |
> > > > | + TDjoint (t=2, m=1)     | 46.8 M | 8.4 B | 224   | 81.6  | 95.8  | + 1.2            | + 0.5            |
> > > > | + TDjoint (t=2, m=1, L4) | 46.0 M | 8.0 B | 224   | 81.1  | 95.5  | + 0.7            | + 0.2            |

---

> > > > > ### Author Response · Authors · 2022-12-12
> > > > > **Response to Point 1 - Empirical Comparison (Part II)**
> > > > >
> > > > > **BANet**
> > > > >
> > > > > The following two tables (Tables 3 and 4) include the object detection and instance segmentation results yielded by BA-Net and MRLA-light with different detectors. Obviously, **MRLA surpasses the BA-Net in nearly all metrics and tasks.**
> > > > >
> > > > > Table 3: Object detection results of different methods using Faster R-CNN and Mask R-CNN
> > > > > as a framework on COCO val2017. $AP^{bb}$ denotes AP of bounding box.
> > > > >
> > > > > | Methods             | Detector    | $AP^{bb}$ | $AP_{50}^{bb}$ | $AP_{75}^{bb}$ | $AP_{S}^{bb}$ | $AP_{M}^{bb}$ | $AP_{L}^{bb}$ |
> > > > > | ------------------- | ----------- | --------- | -------------- | -------------- | ------------- | ------------- | ------------- |
> > > > > | ResNet-50           | Faster-RCNN | 36.4      | 58.2           | 39.2           | 21.8          | 40.0          | 46.2          |
> > > > > | + BA-Net            | Faster-RCNN | 39.5      | 61.3           | 43.0           | **24.5**      | 43.2          | 50.6          |
> > > > > | + MRLA-light (Ours) | Faster-RCNN | **40.4**  | **61.5**       | **44.0**       | 24.2          | **44.1**      | **52.7**      |
> > > > > | ResNet-101          | Faster-RCNN | 38.7      | 60.6           | 41.9           | 22.7          | 43.2          | 50.4          |
> > > > > | + BA-Net            | Faster-RCNN | 41.7      | **63.4**       | 45.1           | 24.9          | 45.8          | 54.0          |
> > > > > | + MRLA-light (Ours) | Faster-RCNN | **42.0**  | 63.1           | **45.7**       | **25.0**      | **45.8**      | **55.4**      |
> > > > > | ResNet-50           | Mask R-CNN  | 37.2      | 58.9           | 40.3           | 23.0          | 43.7          | 51.4          |
> > > > > | + BA-Net            | Mask R-CNN  | 40.5      | 61.7           | 44.2           | 24.5          | 44.3          | 52.1          |
> > > > > | + MRLA-light (Ours) | Mask R-CNN  | **41.2**  | **62.3**       | **45.1**       | **24.8**      | **44.6**      | **53.5**      |
> > > > >
> > > > > Table 4: Instance segmentation results of different methods using Faster R-CNN and Mask R-CNN
> > > > > as a framework on COCO val2017. $AP^{bb}$ denotes AP of bounding box.
> > > > >
> > > > > | Methods             | Detector  | $AP^{m}$ | $AP_{50}^{m}$ | $AP_{75}^{m}$ | $AP_{S}^{m}$ | $AP_{M}^{m}$ | $AP_{L}^{m}$ |
> > > > > | ------------------- | --------- | -------- | ------------- | ------------- | ------------ | ------------ | ------------ |
> > > > > | ResNet-50           | Mask-RCNN | 34.1     | 55.5          | 36.2          | 16.1         | 36.7         | 50.0         |
> > > > > | + BA-Net            | Mask-RCNN | 36.6     | 58.7          | 38.6          | 18.2         | 39.6         | **52.3**     |
> > > > > | + MRLA-light (Ours) | Mask-RCNN | **37.1** | **59.1**      | **39.6**      | **19.5**     | **40.3**     | 52.0         |
> > > > > | ResNet-101          | Mask-RCNN | 35.9     | 57.7          | 38.4          | 16.8         | 39.1         | 53.6         |
> > > > > | + BA-Net            | Mask-RCNN | 38.1     | **60.6**      | 40.4          | 18.7         | 41.5         | **54.8**     |
> > > > > | + MRLA-light (Ours) | Mask-RCNN | **38.4** | **60.6**      | **41.0**      | **20.4**     | **41.7**     | **54.8**     |
> > > > >
> > > > > **ACLA**
> > > > >
> > > > > Since there are no overlapping experiments between ACLA and our paper, we find it difficult to make comparisons with it at this moment.

---

> > > > > > ### Author Response · Authors · 2022-12-12
> > > > > > **Response to Point 2**
> > > > > >
> > > > > > We wish to highlight that our MRLA boosts the performance **by a large margin** in dense prediction tasks. As we mentioned in the empirical evaluation part previously, it even outperforms the latest work, BANet, which is published in ECCV'22.

---

### Official Review · Reviewer_nNZw · 2022-10-22

**Confidence:** 4
**Correctness:** 2
**Technical Novelty And Significance:** 2
**Empirical Novelty And Significance:** Not applicable
**Recommendation:** 6

**Clarity, Quality, Novelty And Reproducibility:**

The paper is clear to read and easy to follow. Although I would appreciate it if the authors provide pseudo codes. The overall quality meets the conference standard. The novelty is not particularly strong. The reproducibility seems ok as the authors provided their codes.

**Strength And Weaknesses:**

The paper presents a network family that combines the attention, densely connected networks, SENet into one unified framework. The overall performance is good, and it’s nice to see this method can work fine with the recent vision transformers given their basic building block is already self-attention. The idea seems simple and straightforward, yet providing performance improvements at little additional costs.

There are some problems about how this paper is formed and how the claims are supported by the experimental evidence.

The first concept I think is `recurrent`. The overall framework could be implemented as a recurrent module. Ignore this comment if I’m wrong but I think the codes show that each building block has its own MRLA instance meaning that the variables are not shared. This leaves a question: does stacking up building blocks make them `recurrent`?

By the way, I think the paper could do better to give us a clearer picture of the underlying implementation by providing some pseudo codes for the two variants in the main paper or appendix. Fortunately, the authors uploaded their codes. So understanding them hasn’t been a problem.

The second problem is the position of the MRLA. The MRLA-light is a self-attention module implemented in the SENet style using sigmoid to replace softmax. It does not look back explicitly, does not compute features recurrently, yet performs similarly as or better than MRLA. This put MRLA in an awkward position that the method following the main topic of the paper more does not show strong advantages over the light variant that follows less.

Finally, I think it will strengthen the paper by showing performance improvements for larger models. The claims are mostly based on experimental results from models at small model scales. Larger model comparisons are necessary for methods that add complexity to the baselines.


**Summary Of The Paper:**

This paper proposes a network family named Multi-head Recurrent Layer Attention (MRLA). It has two module variants: MRLA and MRLA-light. Both of the variants take a self-attention format. A major difference is that MRLA looks back at its previous module’s keys and values, while the light variant only uses the input features for the current layer. The self-attention takes a straightforward format where keys and queries are after global average pooling, and values are pixel features. The modules are tested with both convolutional networks and the vision transformers, and show consistent improvements over the baselines.

**Summary Of The Review:**

The paper proposes a network family for cross-layer attention. The paper is clearly written and the results look solid. What bothers me is that the major topics and claims are not well supported by the network implementation and the experimental evidence.

---

> ### Author Response · Authors · 2022-11-16
> **To Reviewer nNZw (Part II)**
>
> **3. Response to W3: Experiments with larger models**
>
> Thanks for your constructive comments. We have added the experimental results of a larger CNN-based model (ResNet-152) and a larger transformer-based model (DeiT-B) in Table 1 of the main paper. The single-crop accuracies of the baseline models and our MRLA counterpart on the ImageNet-1K validation set are shown below. Note that the results of the original ResNet-152 are copied from the official website of Torchvision.
>
> | Model                    | #Params | #FLOPs | Top-1    | Top-5    |
> | ------------------------ | :------ | ------ | :------- | :------- |
> | ResNet-152 (torchvision) | 60.2 M  | 11.6 B | 78.3     | 94.0     |
> | + SE                     | 66.8 M  | 11.6 B | 78.4     | 94.3     |
> | + ECA                    | 60.2 M  | 11.6 B | 78.9     | 94.6 |
> | + RLA_$g$                | 60.8 M  | 12.3 B | 78.8     | 94.4     |
> | + MRLA-light (Ours)      | 60.7 M  | 11.7 B | **79.1** | **94.6** |
> | DeiT-B                   | 86.4 M  | 16.8 B | 81.8     | 95.6     |
> | + MRLA-light (Ours)      | 86.5 M  | 16.9 B | **82.9** | **96.3** |
>
> References:
>
> 1. Jordan, M. I. (1986). *Serial Order: A Parallel Distributed Processing Approach* (ICS Report 8604). Institute for Cognitive Science, University of California, San Diego .
> 2. Elman, J. L. (1990). Finding structure in time. *Cognitive Science*, 14, 213--252.

---

> > ### Comment · Reviewer_nNZw · 2022-11-30
> > **Acknowledging authors response**
> >
> > Thank you so much for preparing the response to our initial comments in such a short time! The authors have addressed my previous concerns partially, and revised the draft accordingly. Thus, I slightly increased the score to 6. However, as the authors also agreed, there are still some issues about the base vs light as well as the paper's presentation. I also agreed with the other reviewers on the novelty and related works. Considering both the positive and negative comments, I think the paper is slightly above the acceptance bar. Good luck!
> >
> > Cheers

---

> > > ### Author Response · Authors · 2022-11-30
> > > **Thank you for the acknowledgment**
> > >
> > > Thank you very much for acknowledging our efforts in preparing the response and improving the paper. We will try our best to incorporate your current comments into our final version of the manuscript.

---

> ### Author Response · Authors · 2022-11-16
> **To Reviewer nNZw (Part I)**
>
> Thanks very much for your valuable suggestions and comments. We sincerely appreciate your time in evaluating our paper and we have updated our draft following your advice. Our point-to-point responses to your comments are given below.
>
> **1. Response to W1**
>
> **(1) The concept of "recurrent"**
>
> In this paper, we use the concept of "recurrent" based on its definition in recurrent neural networks (RNN), which states that if the output of the network unit depends not only on the input but also on the state of the unit at the previous time step, then the network unit can be considered as recurrent (Jordan, 1986; Elman, 1990).
>
> Correspondingly, in Eq.6, $\boldsymbol{K}^t=\text{Concat}[\boldsymbol{K}^{t-1},f_K^t(\boldsymbol{X}^t)]$ and $\boldsymbol{V}^t=\text{Concat}[\boldsymbol{V}^{t-1},f_V^t(\boldsymbol{X}^t)]$, where the construction of current key and value not only depend on the input $\boldsymbol{X}^t$ but also on the key and value at the previous time step, i.e., $\boldsymbol{K}^{t-1}$ and $\boldsymbol{V}^{t-1}$. Since the key and value in layer attention in Eq.6 are constructed in a recurrent way, we call it recurrent layer attention. Similarly in Eq.8, the output of the $t$-th layer attention $\boldsymbol{O}^t$ not only depends on $\boldsymbol{X}^t$ (through $\boldsymbol{Q}^t$, $\boldsymbol{K}^{t}\_{t,:}$, and $\boldsymbol{V}^t_\{t,:}$) but also on the output of the layer attention at previous time step $t-1$, $\boldsymbol{O}^{t-1}$. Therefore, the resulting layer attention in Eq.8 is also recurrent.
>
> And yes, you are right that in our code, each building block has its own MRLA instance and thus the learning parameters are not shared. However, in MRLA-base, as $\boldsymbol{K}^t$ and $\boldsymbol{V}^t$ inherit from $\boldsymbol{K}^{t-1}$ and $\boldsymbol{V}^{t-1}$, the $t$-th layer attention actually uses the same transformation functions, $f_K^1(\cdot), ..., f_K^{t-1}(\cdot)$ and $f_V^1(\cdot), ..., f_V^{t-1}(\cdot)$, as those used in the $t-1$-th layer attention. The similar idea can also be observed in MRLA-light.
>
> Of course, the above is our understanding and explanation of "recurrent" based on the existing concept in RNNs. And if you still think it is inappropriate, we are ready to make changes to the name of our methods.
>
> **(2) Pseudo codes of MRLA-base and MRLA-light**
>
> Thank you for the helpful suggestion. We have added the pseudo codes of MRLA-base and MRLA-light for implementations in CNNs and vision transformers in Sec. A.5 of the Appendix.
>
> **2. Response to W2: Explanation on MRLA-light**
>
> Thanks for your careful reading and we agree with you that our MRLA-light does not look back explicitly like MRLA-base. Instead, it looks back and performs layer attention in an implicit manner. In the second equality of Eq. 8, we have shown that
> $$
> \boldsymbol{O}^t = \sum\_{l=0}^{t-1} \boldsymbol{\beta}\_l \odot  \left[\boldsymbol{Q}^{t-l} {(\boldsymbol{K}^{t-l}\_{t-l,:})}^{\mathsf{T}} \boldsymbol{V}^{t-l}\_{t-l,:}\right],
> $$
> where $\boldsymbol{\beta}_0=\boldsymbol{1}$, and $\boldsymbol{\beta}_l = \boldsymbol{\lambda}^t_o \odot \cdot\cdot\cdot \odot \boldsymbol{\lambda}^{t-l+1}_o$ for $l\ge1$. The output of MRLA-light is the weighted average of the past layers' features, which implies that MRLA-light indeed retrospectively attends to previous layers.
>
> As for the comparable performances of MRLA-base and MRLA-light, we have to admit that previously we conducted the hyper-parameter tuning and chose the transformation functions $f_Q^t$, $f_K^t$ and $f_V^t$ (where the current choice is GAP + Conv1D and Conv2D) based on MRLA-light as it has a lower computational complexity. We then applied the best combination of hyper-parameters and transformation functions among our attempts to MRLA-base. During the rebuttal period, we have tried another setting for MRLA-base and find that there is a slight increase in its performance (+0.1% in Top-1 single-crop accuracy on the ImageNet-1K validation set).
>
> | Model              | Top-1 acc. | Top-5 acc. |
> | :----------------- | :--------: | :--------: |
> | ResNet-50          |    76.1    |    92.9    |
> | + MRLA-base (Ours) |    77.6    |    93.7    |
>
> Note that the above results of MRLA-base only have a negligible gap with MRLA-light. But, according to our training experiences, it will have a big improvement if we have time to try:
>
> * different kernel sizes for current convolution operations, Conv1D and Conv2D;
> * using transformation functions other than convolutions;
> * more settings for the hyper-parameters such as the number of layer attention heads for each stage;
>
> as what we have done for MRLA-light. However, since MRLA-base has much larger computational complexity, it is very difficult for us to try them all within the short rebuttal period.

---

> ### Author Response · Authors · 2022-11-25
> **Follow-up on response to W2: Updated result of ResNet-50 + MRLA-base**
>
> We would like to update the performance of ResNet-50 + MRLA-base in the image classification task. In this attempt, we change the number of channels per MRLA head from 32 to 16 and add a ReLU activation after the DWConv2D operation. The resulting performance of ResNet-50 + MRLA-base in terms of single-crop accuracy on the ImageNet-1K validation set is as follows.
>
> | Model               | Top-1 acc. | Top-5 acc. |
> | ------------------- | ---------- | ---------- |
> | ResNet-50           | 76.1       | 92.9       |
> | + MRLA-base (Ours)  | **77.7**   | **93.9**   |
> | + MRLA-light (Ours) | **77.7**   | 93.8       |
>
> It can be observed that MRLA-base has a higher Top-5 accuracy and the same Top-1 accuracy as the MRLA-light.
>
> Currently, we are trying to conduct hyper-parameter tuning for EfficientNet-B1 + MRLA-base as its previous setting was also directly borrowed from that of MRLA-light. We believe it will lead to a further improvement on MRLA-base's performance and will keep you updated with our progress.

---

> ### Author Response · Authors · 2022-11-28
> **Follow-up on response to W2: Updated result of EfficientNet-B1 + MRLA-base**
>
> We would like to update you on the encouraging result obtained by EfficientNet-B1 + MRLA-base in the image classification task. Previously, we set the number of channels per MRLA head ($d_k$) to 8 for all stages of EfficientNet-B1 + MRLA-base. In the recent hyper-parameter tuning, we change $d_k$ to 4 for the first five stages since the number of channels is relatively small in these stages. This leads to a higher single-crop accuracy of EfficientNet-B1 + MRLA-base on the ImageNet-1K validation set.
>
> | Model               | Top-1 acc. | Top-5 acc. |
> | ------------------- | ---------- | ---------- |
> | EfficientNet-B1     | 79.1       | 94.4       |
> | + MRLA-base (Ours)  | **80.2**   | **95.3**   |
> | + MRLA-light (Ours) | **80.2**   | 95.2       |

---

### Official Review · Reviewer_XdGN · 2022-10-24

**Confidence:** 4
**Correctness:** 4
**Technical Novelty And Significance:** 3
**Empirical Novelty And Significance:** 3
**Recommendation:** 8

**Clarity, Quality, Novelty And Reproducibility:**

It is straightforward to implement MRLA. And the code is provided. The novelty is minor.


**Strength And Weaknesses:**

Strength
- The core idea is straightforward to implement.
- A wide range of experiments (image classification, object detection, and instance segmentation) are performed on large-scale datasets including ImageNet-1K and COCO. Also, the same module has been tested on quite a few backbone networks including many common variants of ResNet and Vision Transformers.
- The paper is clearly written.

Weakness
- Honestly, I reviewed this paper in the last NeurIPS 2022 and thought the work quality, e.g. solid experiments, well-written,  reached the weak acceptance bar at that time. Given the edited version, the authors have addressed my previous concerns about the experiments. However, my main concerns about this work are still reserved. The novelty of applying layer-wise connection to improve attention is limited, which has been proved by the existing works, e.g, denseNet. And the marginal improvement over existing methods makes it stay slightly above the borderline.
- The ablation study of different layer connections impacting performance is missing.


**Summary Of The Paper:**

This paper presents a module to perform layer attention for CNNs and Vision Transformers in linear-complexity. The core idea is to leverage the representations of previous layers in a recurrent form so that the quadratic-complexity self-attention is avoided. Experiments on image classification, object detection, and instance segmentation are performed to validate the effectiveness of the proposed module.

**Summary Of The Review:**

Though the novelty and performance improvement over SOTA is minor, the work contains solid experiments and comes with good presentation, which provides insights into designing layer-wise connected attention. The idea is straightforward and easy to implement.  Overall, I think it reaches the acceptance bar.

---

> ### Author Response · Authors · 2022-11-14
> **To Reviewer XdGN**
>
> Thanks very much for your acknowledgment of our work's quality and your valuable suggestions. We sincerely appreciate your time in reading the paper, and our point-to-point responses to your comments are given below.
>
> **1.Response to W1**
>
> Thanks for your encouragement. We agree that our objective of strengthening layer interactions through building connections between layers is consistent with that of DenseNet. Nonetheless, our layer attention origins from vanilla attention, which is straightforward yet more effective. In fact, our MRLA outperforms the RLA$_g$-Net, which has been shown to be superior to the DenseNet with similar model sizes. In addition, compared with most existing layer connection methods such as DenseNet, our MRLA can be more easily applied to many state-of-the-art (SOTA) networks without layer connection such that their performances can be significantly improved.
>
> For MRLA's improvements over existing methods, we wish to draw your attention to our comparison strategy. Specifically, we only add our MRLA  module to the backbone network and keep all other parts of the architecture and training strategy unchanged as the original ones. We notice that, in other papers, if the performances on ImageNet are improved by a large margin, they must make large-scale modifications on the network architecture or adopt the advanced training strategy. Furthermore, our improvements in object detection and instance segmentation are especially remarkable compared with other methods.
>
> **2. Response to "the ablation study of different layer connections impacting performance is missing"**
>
> In the following, we are trying to address your concerns since we are not sure about the details on "the ablation study of different layer connections impacting performance". Please let us know if our answers depart from your questions.
>
> We have compared the following layer connections in the ablation study (Sec 5.3):
>
>    * MLA in Eq.4, where we connect two layers via attention but do not reuse previous keys and values (Table 4(a));
>    * MRLA-base,  where we reuse the previous keys and values by concatenation (Table 4(a));
>    * MRLA-light, where the output of the MRLA block for the previous layer is reused by addition (Table 4);
>    * MRLA-light with $\lambda_o^t=1$, which is an identity connection by addition between two layers (Table 4(e));
>    * MRLA-light without $\lambda_o^t O^{t-1}$, which means there is no connection between two layers (Table 4(b)).
>
> Besides, we have also compared MRLA with the following layer connection methods, which were proposed by previous works, in terms of the performances in ImageNet classification, object detection and instance segmentation:
>
>    * RLA_$g$, which connects the layers through a shared RNN unit and not uses attention (Table 1, 2, and 3);
>    * DIANet, which also connects the layers via a shared RNN unit and channel attention (Table 1).
>
> Therefore, if we misunderstood your words, would you mind clarifying them and we are willing to meet your requirements promptly.

---

### Official Review · Reviewer_i4yM · 2022-11-01

**Confidence:** 5
**Correctness:** 3
**Technical Novelty And Significance:** 3
**Empirical Novelty And Significance:** 3
**Recommendation:** 6

**Clarity, Quality, Novelty And Reproducibility:**

The paper is of high quality including its clear motivation, good novelty and originality, and logical presentation.

**Strength And Weaknesses:**

[ Strength ]
+ The motivation is very attractive. It is common that most of existing networks, including Transformers, only focus on the interaction within a certain layer. Even though ResNet and DenseNet, as analyzed by the authors, put some emphasis on layer interaction, the way they used (i.e., addition and/or concatenation) is a little bit naive and hard. The proposed method takes advantage of the attention mechanism and makes the cross-layer interaction more technically rational.
+ The method is elegant and the presentation logic is clear. It is quite straightforward to understand each part of the method section. The proposed efficient implementation of layer attention, just like linear attention in efficient Transformers, is useful and necessary.
+ Sufficient experiments demonstrate the effectiveness of the proposed method. The performance on image classification, object detection and instance segmentation indicates that the method is indeed effective and efficient.

[ Weakness ]
- On image classification, the input resolution is identically set as 224. What if enlarging the resolution? In fact, it is quite important to compare the accuracy and efficiency with various resolutions to reveal the robustness. Of course, it is necessary to check whether the complexity is linear (or nearly linear) to the input resolution.
- Visual comparison is missing. The proposed cross-layer attention can enhance the interaction between layers, but there is no strong evidence specially for this interaction. For example, are local features from shallow layers transferred to top layers? How do high-level features facilitate the feature representation in shallow layers? These would need visual comparison.

**Summary Of The Paper:**

This paper proposes a novel method for cross-layer interaction, which complements current mainstream networks emphasizing the interaction within a layer. Taking advantage of the attention mechanism, the proposed method enhances the layer interaction via attention. An efficient implementation is also introduced to avoid the vanilla quadratic complexity. Experimental results demonstrate the effectiveness of the proposed method.

**Summary Of The Review:**

I think the paper is quite good, and I recommend to accept the paper if the authors can address my two primary concerns above. Also, I would like to have further communication with other reviewers and the AC.

---

> ### Author Response · Authors · 2022-11-17
> **To Reviewer i4yM (Part II)**
>
> **2. Response to W2: Visualization on layer interactions**
>
> Thank you for the constructive comment and we agree that it is necessary to add visualizations to illustrate how layer attention can facilitate the representation learning. In the initial submission, we included the visualizations for DeiT-T (vision transformer) and our MRLA counterpart in Figure 7 of Sec B.5 in Appendix. We now add the visualizations for ResNet-50 (CNN) and MRLA models in Figure 6 of the same sub-section. Please see the highlighted part at the end of our revised Appendix.
>
> In Figure 6, we extract the feature maps from the end of Stage 3 and 4 in ResNet-50 and our MRLA counterparts and visualize them with the score-weighted visual explanations yielded by ScoreCAM (Wang et al., 2020a). Those for Stage 1 and 2 are omitted here as all models have similar feature maps and mainly extract low-level features. The two example images are randomly selected from the ImageNet validation set. In the visualizations, the area with the warmer color contributes more to the classification. It can be observed that:
>
> (1) The models with MRLAs tend to find the critical areas faster than the baseline model. Especially in stage 3, the MRLAs have already moved to emphasize the high-level features while the baseline model still focuses on the lower-level ones;
>
> (2)The areas with the red color in ResNet 50 + MRLA-base/light models are larger than that in the baseline model, implying that the MRLA counterparts utilize more information for the final decision-making;
>
> (3) The patterns of MRLA-base and MRLA-light are similar, validating that our approximation in MRLA-light does not sacrifice too much of its ability.
>
> In Figure 7, we visualize the attention maps of a specified query (red box) from three different heads in the last layer of DeiT-T, DeiT-T + MRLA-base, and DeiT-T + MRLA-light. The first image is randomly sampled from the ImageNet validation set, and the second image is downloaded from a website. In the visualization, the area with the warmer color has a higher attention score. We can observe that MRLA can help the network retrieve more task-related local details compared to the baseline model, indicating that the low-level features are better preserved with layer attention in vision transformers.
>
> One thing we would like to clarify is that our layer attention is autoregressive, i.e., a layer can only attend to previous layers and itself. The $t$-th layer cannot attend to $s$-th layer when $s > t$ as the subsequent layers haven't formed yet. Therefore, the high-level features do not involve the learning of the low-level representations.
>
> Reference:
>
> 1. Haofan Wang, Zifan Wang, Mengnan Du, Fan Yang, Zijian Zhang, Sirui Ding, Piotr Mardziel, and Xia Hu. Score-cam: Score-weighted visual explanations for convolutional neural networks. In *Proceedings of the IEEE/CVF conference on computer vision and pattern recognition workshops*, pp. 24–25, 2020a.

---

> ### Author Response · Authors · 2022-11-17
> **To Reviewer i4yM (Part I)**
>
> Thanks very much for your acknowledgment of our work's quality and your valuable suggestions.
> We sincerely appreciate your time in reading the paper, and our point-to-point responses to your comments are given below.
>
> **1. Response to W1: Experiment with larger resolution**
>
> Following your advice, we add an experiment with CeiT + MRLA-light model, where the input size is 384. The following table shows its single-crop accuracy on the ImageNet-1K validation set, which we have also added to Table 1 of the main paper. In addition, the memory usage and the throughput per GPU are also provided in the table to show MRLA's efficiency.
>
> | Model               | Input | Params | FLOPs | Memory Usage (MiB/GPU) | Throughput per GPU (image/s) | Top-1 | Top-5 |
> | :------------------ | ----- | :----- | :---- | :----------- | -------------------- | :---- | :---- |
> | CeiT-T | 224 | 6.4 M | 1.4 B | 15036 | 1993 | 76.4 | 93.4 |
> | + MRLA-light (Ours) | 224 | 6.4 M (+0.021 M) | 1.4 B (+0.005 B) | 17324 | 1586 | 77.4 | 94.1 |
> | CeiT-T | 384 | 6.4 M | 5.1 B | 17427$^\dagger$ | 656$^\dagger$ | 78.8 | 94.7 |
> | + MRLA-light (Ours) | 384 | 6.4 M (+0.021 M) | 5.1 B (+0.013 B) | 19173$^\dagger$ | 532$^\dagger$ | 79.6 | 95.1 |
>
> $^\dagger$ Note that we use 8 NVIDIA A100 GPUs to compute the memory usage and throughput per GPU for the models with input size 384. For the models with input size 224, we use 4 NVIDIA A100 GPUs to compute these two metrics. To be consistent, we provide the memory usage and throughput per GPU in the table.
>
> To check the complexity induced by MRLA-light with respect to the input resolution. We compute the FLOPs of the baseline CeiT-T and its MRLA counterpart and calculate their differences under various settings of input resolution. It can be observed that the FLOPs induced by MRLA-light are linear to the input resolution. Please see the details in the table below and refer to Figure. 5 in Appendix B.1.2. We have also revised our Sec. 5.1 to include this conclusion.
>
> | Input resolution | FLOPs induced by MRLA-light in CeiT-T |
> | ---------------- | ------------------------------------- |
> | 112 $\times$ 112 | 0.0012 B                              |
> | 128 $\times$ 128 | 0.0015 B                              |
> | 224 $\times$ 224 | 0.0045 B                              |
> | 256 $\times$ 256 | 0.0059 B                              |
> | 384 $\times$ 384 | 0.0133 B                              |
> | 512 $\times$ 512 | 0.0236 B                              |
> | 768 $\times$ 768 | 0.0531 B                              |

---

> > ### Author Response · Authors · 2022-12-05
> > **Additional result of CeiT-S + MRLA-light with large-resolution input**
> >
> > Previously, we supplemented the experiment using CeiT-T + MRLA-light and inputs with larger resolution. Here we would like to provide additional experiment results with larger-resolution inputs and a larger vision transformer model. The following table shows the performance of CeiT-S + MRLA-light in terms of single-crop accuracy on the ImageNet-1K validation set. Consistent improvements brought by our MRLA can be observed.
> >
> > | Model               | Input | Params           | FLOPs             | Top-1 | Top-5 |
> > | :------------------ | ----- | :--------------- | :---------------- | :---- | :---- |
> > | CeiT-S              | 384   | 24.2 M           | 15.90 B           | 83.3  | 96.5  |
> > | + MRLA-light (Ours) | 384   | 24.3 M (+0.06 M) | 15.95 B (+0.05 B) | **84.0**  | **96.9**  |
> >
> > Thank you for the attention and we are really looking forward to receiving your feedback on our updated results.

---

> > > ### Author Response · Authors · 2022-12-07
> > > **Additional result of EfficientNet-B2 + MRLA-light with large-resolution input**
> > >
> > > We would like to update you on the new experiment result with the large-resolution input. This time we examine the effectiveness of our MRLA-light using the EfficientNet-B2 as the backbone and 260 as the input size. The single-crop accuracies on the ImageNet-1K validation set for the baseline and ours are given below. Consistent with previous experiments on the vision transformers, our MRLA is robust in improving the performance.
> > >
> > > | Model               | Input | Params          | FLOPs           | Top-1    | Top-5    |
> > > | :------------------ | ----- | :-------------- | :-------------- | :------- | :------- |
> > > | EfficientNet-B2     | 260   | 9.1 M           | 1.0 B           | 80.1     | 94.9     |
> > > | + MRLA-light (Ours) | 260   | 9.1 M (+0.03 M) | 1.1 B (+0.06 B) | **81.2** | **95.6** |
> > >
> > > As we are approaching the end of the rebuttal period, we would be really grateful if you could kindly let us know whether our responses have addressed your concerns. If you have any further questions, we are more than happy to follow up.

---

> > > > ### Author Response · Authors · 2022-12-12
> > > > **Additional result of DeiT-T + MRLA-light with large-resolution input**
> > > >
> > > > We have conducted the experiment with another vision transformer. The following table shows the performance of DeiT-T + MRLA-light in terms of single-crop accuracy on the ImageNet-1K validation set. Note that the results of the baseline model are reproduced by us. Consistently, adding MRLA leads to improvements in Top-1 and Top-5 accuracies.
> > > >
> > > > | Model               | Input | Params           | FLOPs            | Top-1    | Top-5    |
> > > > | :------------------ | ----- | :--------------- | :--------------- | :------- | :------- |
> > > > | DeiT-T              | 384   | 5.67 M           | 3.15 B           | 74.5     | 92.3     |
> > > > | + MRLA-light (Ours) | 384   | 5.69 M (+0.02 M) | 3.16 B (+0.01 B) | **75.5** | **93.0** |

---

> ### Author Response · Authors · 2022-11-24
> **Summary of our responses**
>
> Thank you again for the time and effort that you dedicated to providing feedback on our paper. We are grateful for your insightful comments and have incorporated them into our updated draft:
>
> 1. We have added the performance of CeiT-T + MRLA-light with the larger input resolution (384$\times$384) in Table 1.
> 2. We have validated the FLOPs induced by our MRLA are nearly linear to the input resolution and included this important observation in Sec 5.1 and Appendix B.1.2.
> 3. We have visualized how MRLA-base and MRLA-light help the representation learning of CNNs and vision transformers in Figures 6 and 7 of Appendix B.5.
>
> The details are included in our previous point-to-point responses. We sincerely hope they can resolve your concerns and welcome further discussion if you have any questions.

---

> ### Author Response · Authors · 2022-12-12
> **We are eager for your feedbacks**
>
> Dear Reviewer i4yM,
>
> It comes to the end of the rebuttal period. Could you please give us a simple comment on our revisions and responses even if you may not be satisfied with them? This is very important for us, say, for another submission if we fail this conference. Many thanks in advance.
>
> Best regards,
>
> ICLR 2023 Conference Paper6346 Authors

---

### Author Response · Authors · 2022-11-18
**To all reviewers**

We sincerely appreciate your reviews and feel that the new revision brought is significantly more complete thanks to them. In the updated version of the paper, changes are marked in **red**. In addition, we address individual comments and questions by commenting on your reviews.

---

### Author Response · Authors · 2022-12-11
**To All Reviewers: Summary of Our Revisions**

Thank you for your time and attention. At the end of the rebuttal period, we would like to provide you with a summary of all our revisions.

1. **Introduction and Related Work**: We have added the most recent works related to layer interactions and attention mechanisms and compared MRLA with them, which further supports the novelty of our paper.
2. **Methodology**:

* The notations of the functions that extract features to derive keys and values in Sec 3.2 have been revised for consistency.
* A detailed explanation of the simplification from Eq. 3 to Eq.5 has been included in Appendix A.2.
* The relationship between our layer attention and techniques that linearize self-attention has been elaborated in Sec 3.3. We have also compared the performance of our MRLA-light with the RLA version that uses another existing linearization technique.
* The assumption made in Eq.7 that query vectors at two consecutive layers have a similar pattern has been verified empirically, where the visualization is in Figure 2(b), and the detailed description is in Sec 5.1.
* Pseudo codes of the MRLA-base and MRLA-light have been supplemented in Appendix A.5.

3. **Experiments**:

* We have added the experiments with larger baseline models, including ResNet-152 and DeiT-B in Table 1.
* We have experimented with large-resolution input with CeiT-T, CeiT-S, and EfficientNet-B2 to show the robustness of our MRLA. Importantly, the FLOPs induced by our MRLA are nearly linear to the input resolution (see Appendix B.1.2).
* A more comprehensive hyper-parameter tuning has been conducted for MRLA-base, further improving its performance in image classification.
*  To investigate how MRLA contributes to the representation learning in CNNs and vision transformers, we have visualized the feature maps and attention maps, respectively, in Appendix B.5.

We really appreciate your constructive comments, as they help us significantly improve the quality of our paper. We sincerely hope these revisions can solve your concerns.

---

### Author Response · Authors · 2023-02-08
**Thanks for your time and support.**

Thanks for your valuable suggestions. Though some reviewers mentioned some relevant works that shared a similar idea, e.g., TDAM (Jaiswal et al., 2022), BANet (Zhao et al. , 2022), and ACLA (Wang et al., 2022b), we would like to bring attention to the fact that we done this work at the beginning of 2022 and submitted our first version to ICML 2022. We searched the literature comprehensively at that time, thus our work and these works can be viewed as **contemporaneous**. Even so, we still would like to include a complete comparison with these most related works.

---

### Decision · Program_Chairs · 2023-01-20

**Decision:**

Accept: poster

**Justification For Why Not Higher Score:**

All reviewers expressed their concerns about the limited novelty of this paper.  A similar idea has already been widely explored in the field, but this paper provided a different implementation. Though the authors mentioned relevant works that share a similar idea, e.g., TDAM (Jaiswal et al., 2022), BANet (Zhao et al. , 2022), and ACLA (Wang et al., 2022b), they only provide a rough comparison at the moment. In the final version, the authors are strongly suggested including a complete comparison with these most related works.

**Justification For Why Not Lower Score:**

All reviewers well recognized the contribution of this paper. Three of them would like to support the acceptance of this paper, while the remaining reviewer expressed his concerns about the out-of-the-date literature review and comparison, which can be addressed in the final version.

**Metareview: Summary, Strengths And Weaknesses:**

This paper introduced a multi-head recurrent layer attention (MRLA) mechanism, the core idea of which is to leverage the representations of previous layers in a recurrent form so that the quadratic-complexity self-attention is avoided.  The authors further proposed a light-weight MRLA to reduce the computational cost. The resulting algorithm can be applied to both CNN and transformer-based networks, and the authors have done basic experiments to evaluate the performance of the proposed algorithms.



**Note From Pc:**

if the above contains the word "oral" or "spotlight" please see: "oral" presentation means -> notable-top-5% and "spotlight" means -> notable-top-25%. As stated in our emails, we are disassociating presentation type from AC recommendations